# Bayesian Image Regression with Soft-thresholded Conditional Autoregressive Prior

**Yuliang Xu**
Department of Statistical Science
Duke University
yuliang.xu@duke.edu

**Jian Kang** [*]
Department of Biostatistics
University of Michigan
jiankang@umich.edu

## Abstract

In the analysis of brain functional MRI (fMRI) data using regression models, Bayesian methods are highly valued for their flexibility and ability to quantify uncertainty. However, these methods face computational challenges in high-dimensional settings typical of brain imaging, and the often pre-specified correlation structures may not accurately capture the true spatial relationships within the brain. To address these issues, we develop a general prior specifically designed for regression models with large-scale imaging data. We introduce the Soft-Thresholded Conditional AutoRegressive (ST-CAR) prior[1], which reduces instability to pre-fixed correlation structures and provides inclusion probabilities to account for the uncertainty in choosing active voxels in the brain. We apply the ST-CAR prior to scalar-on-image (SonI) and image-on-scalar (IonS) regression models—both critical in brain imaging studies—and develop efficient computational algorithms using variational inference (VI) and stochastic subsampling techniques. Simulation studies demonstrate that the ST-CAR prior outperforms existing methods in identifying active brain regions with complex correlation patterns, while our VI algorithms offer superior computational performance. We further validate our approach by applying the ST-CAR to working memory fMRI data from the Adolescent Brain Cognitive Development (ABCD) study, highlighting its effectiveness in practical brain imaging applications.

## 1 Introduction

Regression problems with high-dimensional components have wide-ranging applications. For individual $i$, let a random function $M_i(s), s \in \mathbb{R}^d$ denote the high-dimensional component, and it can be time-series ($d = 1$), 2D images ($d = 2$), or in our particular interest, the functional Magnetic Resonance Imaging (fMRI) data ($d = 3$) of the human brain. In the context of the Adolescent Brain Cognitive Development (ABCD) study (Casey et al., 2018), fMRI data reflects children's brain development. Two scientific problems of interest are (i) the impact of brain development on children's general cognitive ability, formulated as scalar-on-image (SonI) regression; and (ii) the impact of the parental education level on different areas of the children's brain development, formulated as an image-on-scalar (IonS) regression. However, the complex anatomical structure of the human brain, the sparse signal in fMRI data, and the computational burden associated with handling high-resolution 3D images all pose challenges to image regressions.

**Recent Development in Image Regressions** To address these issues, state-of-the-art methods mostly impose sparsity and spatial dependence on the functional parameters. It is well acknowledged in the neuroimaging community that the fMRI data signals are assumed to be sparse and piecewise smooth functions (Kang et al., 2018; Taylor & Worsley, 2007; Smith & Fahrmeir, 2007; Mao et al., 2017). This includes frequentist penalization via total variation distance (Wang et al., 2017), Bayesian Ising priors to control sparsity, and Gaussian Markov Random Field (MRF) or other

---

[*]To whom correspondence should be addressed: Jian Kang (jiankang@umich.edu)

[1]STCAR is available as an R package on Github https://github.com/yuliangxu/STCAR

latent Gaussian Processes to control spatial-correlation (Goldsmith et al., 2014; Huang et al., 2013; Li et al., 2015; Kang et al., 2018; Zeng et al., 2022). A common issue with these approaches is that the spatial smoothness is pre-determined, either through tuning penalty parameters or imposing prior structures on MRF. The data-adaptive way of tuning these smoothness parameters resorts to cross-validation, but with high-resolution 3D fMRI data, cross-validation can be computationally expensive. The T-LoHo method (Lee et al., 2021) proposed a tree-based graph partition prior that can learn spatial correlation treating voxels in 2D space as nodes in a graph, thus avoiding pre-specified smoothness assumption, but this approach is still computationally expensive for 3D fMRI data. In IonS, the popular approach is to use low-rank mapping (Ramsay & Silverman, 2005; Reiss et al., 2010; Yu et al., 2021; Li et al., 2021; Zhu et al., 2014). Again, the choice of the low-rank mapping using basis functions requires fine-tuning to reflect the true correlation structure of the high-dimensional data. Zhang et al. (2023a) proposed to learn the high-dimensional effect through neural networks. Although tuning-free, this only provides point estimation with no uncertainty quantification on the active areas.

**Varitional Inference (VI)** Posterior sampling for high-dimensional large-scale data has been challenging for traditional MCMC methods, especially for imaging applications. As discussed in Blei et al. (2017), VI approximates the posterior sampling problem by an optimization problem, to minimize the Kullback-Leibler (KL) divergence between the posterior density and the candidate density function over a family of densities. This allows us to borrow optimization techniques such as stochastic optimization with subsampling, and develop scalable algorithms for massive imaging data. Under the mean-field assumption, our proposed ST-CAR prior can be easily implemented using Coordinate Ascent VI (CAVI). In addition, inspired by the mini-batch Langevin Dynamic (Wu et al., 2022) algorithm, we also provide a simple modification on CAVI using stochastic subsampling to compute the posterior gradient, which we refer to as Stochastic Subsampling VI (SSVI). One criticism of VI algorithms is that, although VI can provide accurate point estimation, its ability to provide inference and uncertainty quantification on the point estimates is less desirable compared to traditional MCMC algorithms such as the Gibbs sampler. However, by designing a sparsity-inducing prior such as the ST-CAR prior, we can at least get the Posterior Inclusion Probability (PIP) using VIs, which provides statistical confidence whether the effect on certain voxels is nonzero. Downstream FDR control methods (Meyer et al., 2015) can be applied using PIP.

Motivated by the above challenges and computational algorithms, we propose the Soft-Thresholded Conditional AutoRegressive (ST-CAR) prior, which uses a latent variable with CAR structure that is not sensitive to the pre-specified prior correlation and provides shrinkage. Through extensive simulation and real data examples using the ABCD data on both SonI and IonS regressions, we find that ST-CAR with VI algorithms can achieve the best computational performance among all comparable existing methods, and provide top-tier estimation and selection accuracy with state-of-the-art methods. In summary, the proposed ST-CAR prior and its VI algorithms are (i) computationally efficient; (ii) prone to misspecified prior correlation and adaptive to data-driven smoothness; (iii) suitable for active voxels selection and downstream FDR control using PIP; (iv) generalizable not only to SonI and IonS regressions, but also to logistic regression with imaging predictor, or other generalized regression models.

## 2 ST-CAR PRIOR

### 2.1 GENERAL NOTATIONS

Let $\boldsymbol{N}(\mu, \sigma^2)$ represent a normal distribution with mean $\mu$ and variance $\sigma^2$. For the index set $\{1, \ldots, p\}$, let $[-j]$ denote the set $\{1, \ldots, p\} \setminus \{j\}$. For a square matrix $A$, let $\lambda_{\min}(A)$ and $\lambda_{\max}(A)$ be the smallest and the largest eigenvalues of $A$ respectively. Let $C^+(A)$ denote the half Cauchy distribution with density function $f(x) = (2/\pi A)I(x \geq 0)/(1 + x^2/A^2)$. Let $\mathbb{R}^q$ denote the q-dimensional Euclidean space. $I_q$ is the q by q dimensional identity matrix. $\text{IG}(a, b)$ stands for the inverse-gamma distribution. We use $\mathcal{N}(j)$ to denote a neighborhood index set for $j$-th component, and $|\mathcal{N}(j)|$ to denote the cardinality of $\mathcal{N}(j)$. For a vector $\boldsymbol{a}$, $\text{diag}\{\boldsymbol{a}\}$ is used to denote the diagonal matrix with diagonal vector $\boldsymbol{a}$. We use $\text{sgn}(x) = I(x > 0) - I(x < 0)$ to denote the sign of $x$.

## 2.2 ST-CAR PRIOR

The soft-thresholding operator, defined as $T_\nu(x) := \{x - \text{sgn}(x)\nu\}I(|x| > \nu)$ for any $\nu \geq 0$, is a continuous function that shrinks small values below $\nu$ towards zero. The soft-thresholded Gaussian Process (Kang et al., 2018, STGP) prior applies $T_\nu$ to a latent GP. Thus, STGP allows the estimation of functional parameters within the Reproducing Kernel Hilbert Space (RKHS) of the late GP kernel. As the GP kernel and the corresponding RKHS change, the smoothness of the functional parameter space also changes. Without knowing the true signal pattern, pre-specifying the GP kernel can harm STGP's ability to learn the functional parameters (shown in Figure 2). Motivated by this idea, we relax the prior assumption on the smoothness of the functional parameters, and propose a novel noise-relaxed version of the STGP, referred to as the Soft-thresholded conditional auto-regressive (ST-CAR) prior.

**Definition 1.** *A sparse, spatially correlated parameter $\beta(s)$ on a fixed grid $s_1, \ldots, s_p$ follows the ST-CAR prior if*

$$\beta(s_j) \overset{ind}{\sim} \boldsymbol{N}(T_\nu(\mu_j), \sigma_\beta^2), \quad j = 1, \ldots, p$$

$$\mu_j \mid \mu_{\mathcal{N}(j)} \sim \boldsymbol{N}(\bar{\mu}_{\mathcal{N}(j)}, \tau_{\mu,j}^2), \quad \bar{\mu}_{\mathcal{N}(j)} = \rho_j \sum_{k \in \mathcal{N}(j)} b_{j,k}\mu_k$$

*We use $\boldsymbol{\beta} \sim$ ST-CAR$(\nu, B)$ to denote $\boldsymbol{\beta} = (\beta(s_1), \ldots, \beta(s_p))^{\text{T}}$ follow the ST-CAR prior with thresholding parameter $\nu$ and the neighborhood matrix $B$ where $(B)_{j,k} = b_{j,k}$.*

Here, $\beta(\cdot)$ is the target spatially varying parameter. The prior mean of $\beta(s_j)$ is the soft-thresholded $\mu_j$, where $\mu_j$ is a latent parameter for location $s_j$. The prior mean of $\mu_j$ is determined by the correlation coefficient $\rho_j$, the neighborhood set $\mathcal{N}(j)$, and the neighborhood weights $b_{j,k}$.

The variance parameter $\sigma_\beta$ is not identifiable when the ST-CAR prior is applied to SonI or IonS regression. To impose sparsity on $\boldsymbol{\beta}$, we use the annealing idea on $\sigma_\beta$, and let $\sigma_\beta$ decay to 0 as the iteration increases. This will force the spatially independent $\boldsymbol{\beta}$ to converge to the sparse and spatially correlated $\boldsymbol{\mu} = (\mu_1, \ldots, \mu_p)^{\text{T}}$. The decay rate of $\sigma_\beta$ has an impact on the variable selection accuracy especially for low signal-to-noise ratio data. A general rule of thumb is to set $\sigma_\beta$ relatively large at the beginning to allow for more flexibility and decays to a small value at the end.

The correlation parameter $\rho_j$ can either be pre-fixed at all locations, or updated by

$$\rho_j = \delta_j \tilde{\rho}, \quad \delta_j \sim \text{Ber}(p_j),$$

where $\tilde{\rho}_j$ is pre-fixed, and the binary indicator $\delta_j$ help to adaptively determine whether the value of $\mu_j$ is strongly correlated with its neighborhood mean. We find these two approaches of updating or fixing $\rho_j$ have similar performances in variable selection accuracy, and the ST-CAR prior is not sensitive to the choice of $\rho$ or the bandwidth $|\mathcal{N}(j)|$. However, taking the second approach to adaptively update $\boldsymbol{\rho}$ can give us extra information on the correlation structure of the high-dimensional coefficient.

The spatially-correlated structure of the latent $\boldsymbol{\mu}$ is imposed by the Conditional Auto-Regressive (CAR) covariance structure (Gelfand & Vounatsou, 2003). Define a matrix $B$ and a diagonal matrix $D_{\sigma_\mu}$ as

$$(B)_{j,k} = b_{j,k} = \frac{w_{j,k}}{w_{j+}}, \quad (D_{\sigma_\mu})_{j,j} = \tau_{\mu,j} = \frac{\sigma_\mu^2}{w_{j+}} \tag{1}$$

where $w_{j,k}$ are the $(j,k)$-th index of a symmetric matrix $W$, and $w_{j+} = \sum_{k=1}^p w_{j,k}$ is the row(column) summation. The matrix $W$ represents the correlation structure, and in practice we set $w_{j,k} \propto \exp\{-d(s_j, s_k)\}$, exponentially negatively associated with the distance between $s_k$ and $s_j$. In addition to the CAR structure, we set a bandwidth $|\mathcal{N}(j)|$ on the number of components included in the neighborhood $\mathcal{N}(j)$, such that for each $j$, $b_{j,k}$ is nonzero only if $w_{j,k}$ is within the first $|\mathcal{N}(j)|$ largest values among $\{w_{j,k}\}_{k=1}^p$. Denote $\boldsymbol{\rho} = (\rho_1, \ldots, \rho_p)^{\text{T}}$, the joint density of $\boldsymbol{\mu}$ takes the form

$$f(\boldsymbol{\mu}|\sigma_\mu) \propto \exp\left[-\frac{1}{2}\boldsymbol{\mu}^{\text{T}} D_{\sigma_\mu}^{-1} \{I - \text{diag}(\boldsymbol{\rho})B\}\boldsymbol{\mu}\right]$$

If we denote $\Sigma_\mu^{-1} := D_{\sigma_\mu}^{-1}\{I - \text{diag}(\boldsymbol{\rho})B\}$, it can be shown that $\Sigma_\mu^{-1}$ is a symmetric, positive definite matrix when each $\rho_j \in \left(\lambda_{\min}(B)^{-1}, \lambda_{\max}(B)^{-1}\right)$. Note that $B$ is not a symmetric matrix in general. In the construction in equation 1, $\lambda_{\max}(B) = 1$ and $\lambda_{\min}(B) < 0$. Hence we choose $\rho_j \in [0, 1)$ for any $j$, and the joint density of $\boldsymbol{\mu}$ is guaranteed to be non-degenerative. One caveat of doing so is that the value of $\mu_j$ and the neighborhood mean $\bar{\mu}_{\mathcal{N}(j)}$ is only allowed to be either positively correlated or independent ($\rho_j \geq 0$), but the negative correlation is not taken into consideration. This constraint makes sense in brain imaging applications, because the true signal is assumed to be sparse and piecewise smooth, which excludes the case where the signal across neighboring voxels has a sharp drop from positive to negative values. In general, the ST-CAR prior is suitable for the case where the positive and negative areas do not share boundaries.

The proposed ST-CAR prior enjoys good computational properties as it has a conditional conjugate posterior when applied to a spatial parameter in a regression problem. The main challenge in updating a thresholded parameter is that the thresholding function such as $T_\nu$ is non-linear. But we can show that $\mu_j$ conditional on all other $\mu_{[-j]}$ and $\beta$ has a mixture of truncated normal distribution as its posterior.

**Proposition 1.** *Within the ST-CAR prior, the posterior of $\mu_j$ can be expressed as a mixture of three truncated normal distributions.*

$$\pi(\mu_j \mid \boldsymbol{\beta}, \mu_{[-j]}, \sigma_\mu, \sigma_\beta) =$$
$$P_j^+ \cdot \boldsymbol{N}_{[\nu,+\infty)}(\mu_j^+, V_j) + P_j^0 \cdot \boldsymbol{N}_{[-\nu,\nu]}(\bar{\mu}_{\mathcal{N}(j)}, V_0) + P_j^- \cdot \boldsymbol{N}_{(-\infty,-\nu]}(\mu_j^-, V_j) \tag{2}$$

The expression for $P_j^+, P_j^0, P_j^-, \mu_j^+, \mu_j^-, V_j, V_0$ can be found in the proof of Proposition 1 in the Appendix. Proposition 1 provides the closed-form posterior density for the sparse-mean latent parameter $\{\mu_j\}_{j=1}^p$ as a mixture of 3 truncated normal distributions in the ST-CAR prior. Due to the normal mixture conjugacy, the Gibbs sampler can be directly applied. Proposition 1 and the prior property $\mathbb{E}(\beta_j|\mu_j = 0) = 0$ will allow us to approximate the posterior inclusion probability using $(1 - P_j^0)$ given data.

The proposed ST-CAR prior is a general prior that can be applied to many high-dimensional regression settings, where the coefficient is assumed to be smooth and sparse across their spatial domain. SonI and IonS regressions in Section 2.3 are two examples to illustrate the power of ST-CAR prior. Other potential applications include logistic regression with high-dimensional exposure and other types of generalized linear models.

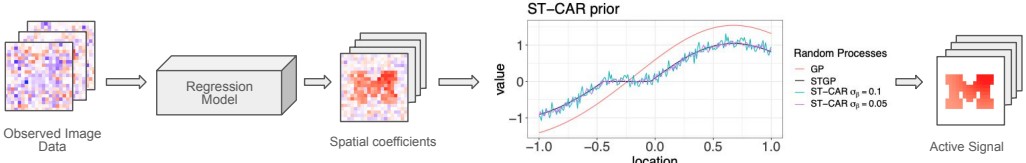

Figure 1: Illustration to use ST-CAR prior for regression models with imaging data.

## 2.3 APPLICATION TO IMAGE REGRESSION MODELS

The ST-CAR prior can be applied to various models with sparse and spatially varying functional parameters as shown in Figure 1. This section demonstrates its advantage using SonI and IonS models. We focus on the SonI model and defer IonS model details to the Appendix.

**Scalar-on-Image (SonI) Regression** Let $M_i(s_j)$ denote the fMRI signal intensity at location $s_j$ for individual $i$, $\mathbf{X}_i \in \mathbb{R}^q$ be a vector-valued confounder variables. Let $Y_i$ denote the scalar-valued outcome for subject $i$. $i = 1, \ldots, n, j = 1, \ldots, p$.

$$Y_i = \sum_{j=1}^p \beta(s_j)M_i(s_j) + \boldsymbol{\gamma}^{\mathrm{T}}\mathbf{X}_i + \epsilon_i \quad \epsilon_i \stackrel{\text{iid}}{\sim} \boldsymbol{N}(0, \sigma_Y^2)$$
$$\boldsymbol{\beta} \sim \text{ST-CAR}(\nu, B) \tag{3}$$

where $\boldsymbol{\beta} \in \mathbb{R}^p$ is the high-dimensional spatially-varying coefficient of interest, and $\boldsymbol{\gamma} \in \mathbb{R}^q$ is the vector-valued coefficient for the confounders $\mathbf{X}_i$.

The remaining parameters are assigned the following priors $\boldsymbol{\gamma} \sim \boldsymbol{N}(0, \sigma_\gamma^2 I_q), \sigma_Y \sim C^+(1), \sigma_\gamma \sim C^+(1)$. For all of the Half-Cauchy parameters, we use their equivalent conjugate form to update: $\sigma_Y \sim C^+(1)$ is equivalent to $\sigma_Y^2 \sim \text{IG}(1/2, 1/a_Y), a_Y \sim \text{IG}(1/2, 1)$. Because $\boldsymbol{\beta} \sim \text{ST-CAR}(\nu, B)$ essentially assigns spatially independent prior to $\boldsymbol{\beta}$ with a latent spatial-correlated mean function, we can use singular value decomposition (SVD) on the design matrix $M \in \mathbb{R}^{n \times p}$ to boost further the computation speed[2].

**Image-on-Scalar (IonS) Regression** In IonS regression, let the spatially varying outcome be the fMRI data $M_i(s_j)$. The exposure of interest is denoted as $X_i$, and the confounders are denoted as $\mathbf{C}_i \in \mathbb{R}^m$. The IonS model is as follows

$$M_i(s_j) = \alpha(s_j)X_i + \sum_{k=1}^m \xi_k(s_j)C_{i,k} + \eta_i(s_j) + \epsilon_{i,j}, \quad \epsilon_{i,j} \overset{\text{iid}}{\sim} N(0, \sigma_M^2)$$

$$\boldsymbol{\alpha} \sim \text{ST-CAR}(\nu, B), \tag{4}$$

with other prior specification as follows: $\xi_k \overset{\text{iid}}{\sim} \text{GP}(0, \sigma_\xi^2 \kappa), k = 1, \ldots, m, \quad \eta_i \overset{\text{iid}}{\sim} \text{GP}(0, \sigma_\eta^2 \kappa), \sigma_M \sim C^+(1), \quad \sigma_\xi \sim C^+(1), \quad \sigma_\eta \sim C^+(1)$. Here, we only assign ST-CAR to $\alpha$ for selecting active regions for the exposure. We defer details of IonS to Appendix A.1.

## 3 VARIATIONAL INFERENCE ALGORITHMS

This section uses the SonI equation 3 as an example, and introduces variational inference algorithms: Coordinate Ascent Variational Inference (CAVI) and Stochastic subsampling version of variational inference (SSVI). More computational details can be found in A.2 and A.3.

The variational inference methods (CAVI, SSVI) are based on the mean-field assumption (Blei et al., 2017). If we denote $\theta = (\boldsymbol{\beta}, \boldsymbol{\gamma}, \boldsymbol{\mu}, \sigma_Y, \sigma_\gamma)$ as the collection of all parameters. The mean-field variational inference minimizes the evidence lower bound

$$\min_q \mathbb{E}\left[KL(q(\theta) \mid p(\theta|\mathbf{Y}, \mathbf{M}, \mathbf{X}))\right] \quad s.t. \ q(\theta) = q(\boldsymbol{\beta})q(\boldsymbol{\gamma})q(\boldsymbol{\mu})q(\sigma_Y)q(\sigma_\gamma).$$

**Coordinate Ascent Variational Inference (CAVI)** The classic CAVI algorithm iteratively refines the approximated density $q$ by updating each parameter in successive iterations $t$ by the following density approximation,

$$\log q^{(t)}(\boldsymbol{\beta}) \propto \mathbb{E}_{q^{(t-1)}(\boldsymbol{\gamma}, \boldsymbol{\mu}, \sigma_Y, \sigma_\gamma)} \left\{\log p(\boldsymbol{\beta} \mid \mathbf{Y}, \mathbf{M}, \mathbf{X}, \boldsymbol{\gamma}, \boldsymbol{\mu}, \sigma_Y, \sigma_\gamma)\right\}.$$

Because each prior in SonI has closed-form posterior density, we can directly apply this iterative approach.

One issue with the conventional CAVI is that although it can give a good point estimation as an optimization algorithm, but cannot directly make inferences such as the credible interval, compared with MCMC sampling methods. The novelty in our proposed ST-CAR prior is that we can use the mixing probability in Proposition 1 as the uncertainty quantification measure for selecting significant regions, circumventing the requirement for credible interval based on MCMC samples, while leveraging the computational efficiency provided by CAVI. Proposition 1 gives the posterior probability of $\mu_j$ belonging to the positive group $[\nu, \infty)$, zero group $[-\nu, \nu)$, and negative group $(-\infty, -\nu)$. When using CAVI, we can directly compute the Posterior Inclusion Probability (PIP) under $q$ density as $(P_j^+ + P_j^-)$ in equation 2 as a measure of coefficient significance.

**Stochastic Subsampling Variational Inference (SSVI)** The idea of SSVI is similar to the Stochastic Gradient Langevin Dynamics (Welling & Teh, 2011, SGLD), in the spirit that instead of using the entire sample, we take a subsample of data, indexed by $I \subset \{1, \ldots, n\}$, and apply stochastic gradient updating algorithm. In the context of VI update of $\beta_j$, let $s_t$ be the step size at $t$-th iteration,

---

[2]Details on this derivation can be found in the Appendix

let $\pi$ be the prior density of $\beta_j$,

$$\mathbb{E}_{q^{(t)}}\{\beta_j\} \leftarrow \mathbb{E}_{q^{(t-1)}}\{\beta_j\} + s_t \left( \frac{n}{n_s} \nabla \mathbb{E}_{q^{(t-1)}} \log \sum_{i \in I} p(Y_i, \mathbf{M}_i, \mathbf{X}_i \mid \theta) + \nabla \mathbb{E}_{q^{(t-1)}} \log \pi(\beta_j) \right).$$

We defer details of SSVI to Section A.3. SSVI is particularly useful for large-scale problems such as IonS where the likelihood computation is expensive due to the high-dimensional outcome, whereas the performances of SSVI and CAVI for SonI are similar.

## 4 NUMERICAL EXAMPLES

In this section, we focus on simulation for SonI equation 3, and defer simulation results on IonS equation 4 regressions to A.4. Our primary goal is to compare the proposed prior ST-CAR with other existing methods using CAVI. Because CAVI offers a balance between computational efficiency and selection accuracy in SonI setting. Empirical comparisons of the posterior computation algorithms including Gibbs sampler, SSVI, and CAVI are included in A.10.

**Simulation I: Scalar-on-image regression with CAVI**

For SonI model equation 3, we compare ST-CAR with 3 other methods: (1) Soft-thresholding Gaussian Process prior (STGP) Kang et al. (2018), (2) T-LoHo Lee et al. (2021), (3) Elastic Net Zou & Hastie (2005).

For the elastic net implemented in the glmnet package (Friedman et al., 2010a), the mixing parameter $\alpha$ is set to 0.5, and the penalty parameter $\lambda$ is chosen using cross-validation.

The STGP prior is based on soft-thresholding on the latent Gaussian Process. When $\beta(s) \sim \mathcal{STGP}(\nu, \kappa)$, there exists a corresponding latent Gaussian Process $\tilde{\beta}(s) \sim \mathcal{GP}(0, \kappa)$ such that $\beta(s) = T_\nu(\tilde{\beta})$. This method requires a pre-specified kernel function $\kappa$, and the posterior sampling algorithm is the Metropolis-adjusted Langevin algorithm (MALA). In this simulation, we use the exponential square kernel

$$\kappa(s, s'; a, b) = \text{cor}\{\beta(s), \beta(s')\} = \exp\{-a(s^2 + s'^2) - b(s - s')^2\} \tag{5}$$

where $a = 0.01, b = 10$. The implementation is based on BIMA package [3] (Xu & Kang, 2023). Note that this implementation of STGP allows the users to specify different regions in the image and specify a region-wise independent kernel in order to speed up the computation in high dimensions and boost selection accuracy in each region. Hence for the simulation pattern shown in Figure 2, we evenly split the entire 2D region into 4 sub-regions, and use the modified exponential square kernel on each sub-region. The basis function is generated using Kang (2022) with 10 degrees of Hermite polynomials for each sub-region. We use the elastic net result as the initial value for $\boldsymbol{\beta}$, and run a total of $10^4$ iterations with the last 20% as the converged MCMC sample. The thresholding parameter $\nu$ is set to be 0.2. For the variable selection accuracy, we use the Posterior Inclusion Probability (PIP) based on the MCMC sample of $\boldsymbol{\beta}$, defined as $\text{PIP}_j = \sum_{t=1}^{T} I(\beta_j \neq 0)/T$ for the location $j$ with $T$ MCMC sample.

The T-LoHo [4] method is designed for clustering nodes in graph models into finite discrete values, and it shows great performance for this purpose especially under low SNR. However, it is not well suited to capture continuous functional parameters. In addition, T-LoHo does not impose sparsity. We use the 95% credible interval to select active voxels. T-LoHo package does not provide the confounder coefficients estimation, hence we set true $\boldsymbol{\gamma} = 0$. We use a total of 50000 MCMC iterations and take the last 10000 as the converged sample.

For ST-CAR prior updated using the CAVI algorithm, we use the ridge regression result as the initial value for $\boldsymbol{\beta}$, and set the initial value for $\boldsymbol{\mu}$ to be all 0. The thresholding parameter $\nu$ is set to be the largest marginal value in $\boldsymbol{\beta}$ estimated from ridge regression. This is because setting $\nu$ to be a large value can reduce false discoveries, and $\boldsymbol{\mu}$ is still able to recover the true signal pattern even when starting from all 0 initial values. This algorithm is much less sensitive to the thresholding parameter

---

[3]BIMA package https://github.com/yuliangxu/BIMA
[4]TLOHO package https://github.com/changwoo-lee/TLOHO

compared to STGP. The decay rate of $\sigma_\beta^2$ is set to be $0.5(1+t)^{-0.7}$ where $t$ represents the number of iterations. Since we use annealing on $\sigma_\beta^2$ instead of fully conjugate update, we can no longer use ELBO as a stopping rule. Instead, at the $t+1$ iteration, we compute the difference of $\boldsymbol{\beta}^{(t)}$ and $\boldsymbol{\beta}^{(t+1)}$, defined as $\sum_{j=1}^p (\beta_j^{(t)} - \beta_j^{(t+1)})^2/p$, to determine whether the optimization has converged. The tolerance is set to be $10^{-10}$. For the neighboring matrix $B$, we set the number of neighbors as 8, and the correlation parameter $\tilde{\rho}$ is set to be 0.9. The variance parameter $\sigma_\mu$ is fixed at 1 for the CAVI update.

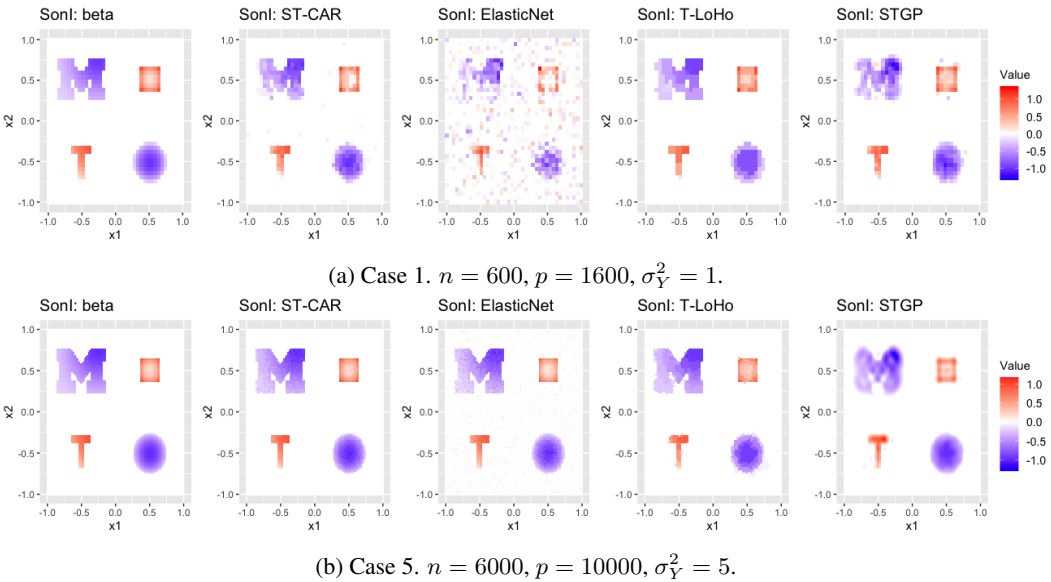

(a) Case 1. $n = 600$, $p = 1600$, $\sigma_Y^2 = 1$.

(b) Case 5. $n = 6000$, $p = 10000$, $\sigma_Y^2 = 5$.

Figure 2: SonI result illustration for all competing methods. The first figure in each row is the true $\beta$ signal.

Figure 2a and 2b provide visual comparison under two simulation settings. The true $\boldsymbol{\beta}$ image is designed to include several challenging patterns where the active area can decay smoothly to almost 0, to have a complex correlation structure such as the M-shape on the top-left corner, and to include both positive and negative patterns. Case 1 (Figure 2a) is the low-resolution and low SNR setting, and Case 5 (Figure 2b) is the high-resolution and high SNR setting.

From this visual comparison, STGP is good at estimating smooth function patterns such as the bottom-left circle, but without further tuning the Gaussian Process kernel, estimating more complex patterns such as the M-shape would be difficult. Here, STGP already considers the region partition. If we were to generate a Gaussian Kernel over the entire support, the result would be even smoother (appear more blurry on sharp edges). T-LoHo as a clustering method is good at grouping larger effects, but as the true signal decays smoothly towards 0, T-LoHo can ignore some small non-zero effects, resulting in a lower statistical power, as shown in the bottom-left circle in Figure 2b. Elastic Net can identify the spatial pattern to a certain extent, but the power is limited as it does not leverage spatial information. Consequently, it may yield a noisy estimation in case 1. Even in case 5, where the point estimation is favorable, Elastic Net can still introduce background noises. ST-CAR can estimate each pattern relatively well without specifying any region partition or tuning the correlation matrix and it automatically adapts to different signal patterns. Although some small effects such as the bottom tip of the T-shape can still be missed, ST-CAR provides the best overall performance compared to other priors across different settings without any tuning procedure.

Table 1 provides a detailed numerical comparison. The evaluation criteria for estimation accuracy include (i) Selection accuracy: false discovery rate (FDR), true positive rate (TPR), and overall accuracy (ACC); (ii) Point estimation: root mean squared error (RMSE); (iii) Goodness-of-fit: the predictive mean squared error on the outcome $Y_i$ using training and testing data (train and test pMSE). We also include the computational time comparison averaged over 100 replications. Because CAVI is an optimization algorithm, we can set a stopping rule, whereas for MCMC sampling algorithms

Table 1: Numeric result for SonI simulation, under 100 replications.

(a) SonI: Comparison of estimation accuracy. The evaluation criteria for estimation includes false discovery rate (FDR), true positive rate (TPR), overall accuracy (ACC), and root mean squared error (RMSE), all multiplied by 100. The evaluation criteria for predictive performance includes training and testing predictive MSE, denoted as Train and Test pMSE respectively

| | Case 1. $n=600, p=1600, \sigma_Y^2=1$ | | | | | Case 2. $n=600, \boldsymbol{p=900}, \sigma_Y^2=1$ | | | |
| --- | --- | --- | --- | --- | --- | --- | --- | --- | --- |
| | ST-CAR | ElasNet | STGP | T-LoHo | | ST-CAR | ElasNet | STGP | T-LoHo |
| FDR | 9.52 | 9.54 | 10.42 | 4.41 | FDR | 9.40 | 1.82 | 9.77 | 3.92 |
| TPR | 97.47 | 44.85 | 95.37 | 97.32 | TPR | 99.81 | 98.45 | 94.06 | 99.34 |
| ACC | 98.05 | 90.87 | 97.60 | 98.62 | ACC | 98.45 | 99.50 | 97.63 | 98.65 |
| RMSE | 11.00 | 30.45 | 13.44 | 9.46 | RMSE | 5.51 | 7.12 | 13.00 | 6.56 |
| Train pMSE | 2.29 | 1.03 | 5.97 | 3.61 | Train pMSE | 1.10 | 0.33 | 3.40 | 1.55 |
| Test pMSE | 7.18 | 60.31 | 12.54 | **6.21** | Test pMSE | **2.02** | 3.04 | 6.36 | 2.24 |
| | Case 3. $\boldsymbol{n=1000}, p=1600, \sigma_Y^2=1$ | | | | | Case 4. $n=600, p=1600, \boldsymbol{\sigma_Y^2=5}$ | | | |
| | ST-CAR | ElasNet | STGP | T-LoHo | | ST-CAR | ElasNet | STGP | T-LoHo |
| FDR | 9.39 | 0.39 | 9.89 | 1.10 | FDR | 9.64 | 9.42 | 9.83 | 7.09 |
| TPR | 100.00 | 99.08 | 97.93 | 99.79 | TPR | 90.06 | 37.11 | 92.00 | 94.19 |
| ACC | 98.42 | 99.80 | 98.05 | 99.76 | ACC | 97.02 | 89.82 | 97.25 | 97.09 |
| RMSE | 4.27 | 6.53 | 12.04 | 5.83 | RMSE | 14.65 | 33.31 | 14.72 | 12.51 |
| Train pMSE | 1.08 | 0.48 | 6.47 | 1.92 | Train pMSE | 4.85 | 2.55 | 9.17 | 8.56 |
| Test pMSE | **2.03** | 3.79 | 9.92 | 2.78 | Test pMSE | 17.76 | 76.84 | 19.31 | **14.37** |
| | Case 5. $n=6000, p=10000, \boldsymbol{\sigma_Y^2=5}$ | | | | | Case 6. $n=6000, p=10000, \boldsymbol{\sigma_Y^2=10}$ | | | |
| | ST-CAR | ElasNet | STGP | T-LoHo | | ST-CAR | ElasNet | STGP | T-LoHo |
| FDR | 0.46 | 0.27 | 17.16 | 6.84 | FDR | 1.18 | 2.47 | 17.15 | 4.78 |
| TPR | 99.98 | 99.39 | 98.52 | 99.67 | TPR | 99.91 | 98.21 | 98.53 | 99.52 |
| ACC | 99.92 | 99.86 | 96.57 | 95.96 | ACC | 99.80 | 99.33 | 96.58 | 97.71 |
| RMSE | 3.73 | 6.73 | 13.78 | 5.25 | RMSE | 3.14 | 3.94 | 6.44 | 2.48 |
| Train pMSE | 5.48 | 3.12 | 80.90 | 12.78 | Train pMSE | 10.62 | 3.81 | 85.91 | 18.60 |
| Test pMSE | **9.37** | 22.94 | 86.88 | 15.54 | Test pMSE | **18.43** | 41.06 | 92.25 | 22.44 |

(b) Computation time for SonI simulation, averaged over 100 replications.

| Computation time | Total time (seconds) | | | Number of iteratios per second | | |
| --- | --- | --- | --- | --- | --- | --- |
| Case | ST-CAR | STGP | T-LoHo | ST-CAR | STGP | T-LoHo |
| Case 1. $n=600, p=1600, \sigma_Y^2=1$ | 103.0 | 503.0 | 306.5 | 11.4 | 208.0 | 262.6 |
| Case 2. $n=600, \boldsymbol{p=900}, \sigma_Y^2=1$ | 24.2 | 250.8 | 205.0 | 42.3 | 420.5 | 393.4 |
| Case 3. $\boldsymbol{n=1000}, p=1600, \sigma_Y^2=1$ | 111.2 | 866.2 | 426.2 | 10.2 | 122.9 | 189.5 |
| Case 4. $n=600, p=1600, \boldsymbol{\sigma_Y^2=5}$ | 108.5 | 486.0 | 312.5 | 11.1 | 212.4 | 259.8 |
| Case 5. $\boldsymbol{n=6000, p=10000, \sigma_Y^2=5}$ | 8034.9 | 40658.8 | 11141.1 | 0.2 | 2.6 | 7.3 |
| Case 6. $\boldsymbol{n=6000, p=10000, \sigma_Y^2=10}$ | 7811.3 | 40839.1 | 11297.8 | 0.2 | 2.6 | 7.2 |

for STGP and T-LoHo, a lot more iterations are required. Hence we report both the total time and the number of iterations per second.

For the variable selection results for ST-CAR, Elastic Net, and STGP, we use a tuning procedure to find a cutoff such that the FDR can be controlled below 10% within a fixed tuning window. For STGP and ST-CAR, the PIP is used to control FDR. For elastic net, effect size of $\boldsymbol{\beta}$ is used to control FDR. For T-LoHo, the 95% CI is used.

Based on Table 1, we can see that ST-CAR has the lowest testing pMSE in 3 relatively high SNR cases (Case 2,3,5). For Case 1 and 4 with relatively low SNR, ST-CAR has the second-best performance next to T-LoHo. For the computation time in Table 1b, ST-CAR has the best computational efficiency across all cases.

## 5 APPLICATION TO ABCD STUDY

In this section, we apply ST-CAR to analyze the Adolescent Brain Cognitive Development (ABCD) study release 1 data (Casey et al., 2018). The ABCD study is a long-term study on the brain development of children in the United States. In this real data analysis, we use the 2-back 3mm task fMRI contrast data (Sripada et al., 2020a). The preprocessing and the general cognitive score (g-score) follow from the previous study Sripada et al. (2020b). The scientific questions of interest are: (i) whether the brain signals in different regions have different impacts on the children's IQ score

(SonI); (ii) whether parents with higher education degrees has an impact on the children's cognitive ability development (IonS). For the task fMRI data, after preprocessing (Sripada et al., 2020b) and removing subjects with missing covariates, we have $p = 47636$ voxels and $n = 1861$ subjects in total.

To answer (i) with SonI model equation 3, we use the children's IQ score as the scalar outcome $Y_i$, and use the task fMRI data as the high-dimensional predictor $M_i(s_j)$, where $s_j$ stands for voxel locations in the brain. The confounders include parental education level (binary, 1 if the parent has a bachelor's degree or higher), age, gender, race and ethnicity (Asian, Black, Hispanic, Other, White), and household income (less than 50k, between 50k and 100k, greater than 100k). The coefficient of interest is $\beta$ in equation 3. We expect $\beta$ to be very sparse and have small effects, since the interpretation for $\beta(s_j) = b$ is that one unit increase in the brain signal in location $s_j$ is associated with $b$ amount of change in the children's IQ score, and the range of the standardized IQ score is $(-2.84, 3.26)$, a small range compared to a large number of predictors $p = 47636$.

To answer (ii) with the IonS model equation 4, we use the task fMRI data as the outcome, and the parental education level as the exposure. The confounders include age, gender, race and ethnicity, and household income. For the IonS model, for $\xi_k, \eta_i$ that are assigned GP priors. The interpretation for $\alpha(s_j) = a$ in equation 4 is that parents with bachelor's degrees or higher are associated with $a$ amount of change in the brain signal at location $s_j$. Hence we expect the effect size of $\alpha$ to be relatively larger than that of $\beta$.

In ST-CAR prior, the two most important tuning parameters are the thresholding parameter $\nu$, and the initial value for $\sigma_\beta^2$ which controls how close $\beta$ is to the latent sparse $\mu$. In theory (Kang et al., 2018), the choice of $\nu$ does not have a huge impact as long as the initial values are close enough to the truth, or the MCMC sampling algorithm can run long enough to fully explore the parameter space. Because we are using VI algorithms, it is important to start with a good initial value. Hence we perform a sensitivity analysis to select the best $\nu$ and initial $\sigma_\beta^2$ in terms of the smallest testing pMSE. The entire data set is split into 70% training data and 30% testing data. Based on the sensitivity analysis results in Table A2 and Appendix Table A4 and A3, we choose $\nu = 0.007$, the initial value for $\sigma_\beta^2$ to be $10^{-5}$, bandwidth 9 and decay rate $\gamma = 0.35$ in the decay rate function of $\sigma_\beta^2$ for SonI, and choose $\nu = 0.005$, the initial value for $\sigma_\alpha^2$ to be 0.1, bandwidth 26 and decay rate $\gamma = 0.45$ for IonS. Although varying $\nu$, bandwidth and decay rate have little influence on the results. Table A2 also reflects that our method has better testing pMSE compared to the competing methods Elastic Net, STGP, and MUA. Due to the computational limitation of other methods, we chose not to run all competing methods on the real data.

We use CAVI on SonI, which takes 1.7 hours to run, and SSVI on IonS, which takes 7.3 hours to run. Due to the vast sparsity and low SNR in $\beta$, the computational time of SonI is similar to STGP (1.6 hours). However, the IonS model with the SSVI algorithm shows a huge computational improvement compared to STGP (85.9 hours).

We present the final data analysis result in both visual illustrations in Figure 3, and numeric values in Table 2. Figure 3 is a visualization of the positive significant voxels in SonI and IonS. The color range for the plots is between $[a, b]$, where only voxels with values greater than $a$ are shown, and voxels with values greater than $b$ are shown in the brightest color. From Figure 3a, due to the low SNR in SonI, both the effect size and PIP are small, and only a small amount of voxels with a large effect size aligns with the mapping of PIP greater than 0.25. In comparison, $\alpha$ in IonS has a larger effect size, and as shown in Figure 3b, the large effect areas align well with the mapping of PIP greater than 0.98.

In Table 2, we show the region-level numeric result. Note that, although both SonI and IonS have a small amount of negative effects, they are very close to 0 compared to the positive effect scale, hence we only report the positive effect here. From Table 2, for SonI, *Precuneus_L* is the region with the largest positive effect, which means brain development in this region can have the most positive effect on the children's IQ score. This aligns with the previous study in Xu & Kang (2023) and scientific findings (Wallentin et al., 2006) that Precuneus is related to memory tasks. For IonS, *Frontal_Mid* region in both the left and right hemispheres have the largest positive effect, and have been shown to play a key role in the development of literacy (left *Frontal_Mid*) and numeracy (right *Frontal_Mid*) in previous findings (El-Baba & Schury, 2020).

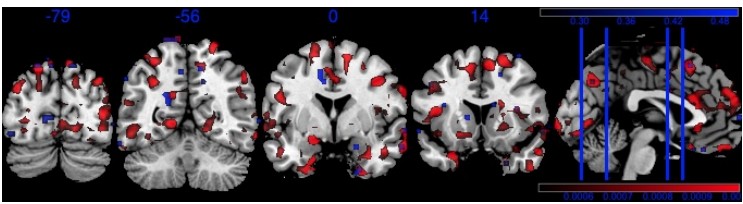

(a) SonI: values of $\beta$ (Red) w. color range [0.0005,0.001], and values of PIP (overlaying blue) w. color range [0.25, 0.5].

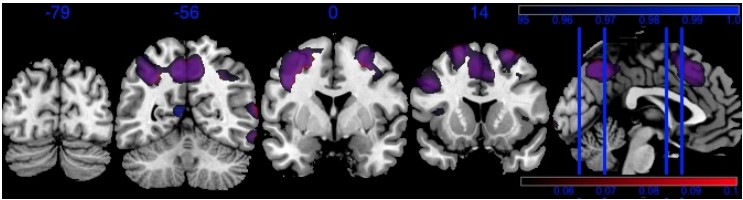

(b) IonS: values of $\alpha$ (Red) w. color range [0.05,0.1], and values of PIP (overlaying blue) w. color range [0.95,1].

Figure 3: Visual illustration of $\beta$ in SonI and $\alpha$ in IonS.

Table 2: Numeric result for the top 10 regions sorted by number of significant positive voxels in SonI and IonS. For SonI, *sig count* is the number of significant voxels (PIP$_j \geq 0.25$) in each region, *pos_sig count* is the number of significant voxels with $\beta(s_j) \geq 0.0005$, and *pos sum* is $\sum_{j \in \mathcal{S}_r} \beta(s_j) I(\beta(s_j) > 0)$, the sum of positive effect for all voxels in region $r$. The IonS result has the same interpretation, except the cutoff for significant voxels is PIP$_j \geq 0.95$, and the cutoff for positive effect in *pos_sig count* is 0.05.

| | SonI | | | | | IonS | | | |
|---|---|---|---|---|---|---|---|---|---|
| region name | region code | sig count | pos_sig count | pos sum | region name | region code | sig count | pos_sig count | pos sum |
| Precuneus_L | 67 | 12 | 12 | 0.25 | Parietal_Inf_L | 61 | 382 | 357 | 38.94 |
| Temporal_Sup_R | 82 | 12 | 9 | 0.16 | Precuneus_L | 67 | 377 | 312 | 37.26 |
| Temporal_Inf_R | 90 | 18 | 9 | 0.18 | Precentral_L | 1 | 305 | 293 | 33.55 |
| Precuneus_R | 68 | 12 | 8 | 0.14 | Precuneus_R | 68 | 316 | 285 | 36.24 |
| Temporal_Inf_L | 89 | 14 | 8 | 0.15 | Frontal_Mid_R | 8 | 322 | 270 | 43.12 |
| Occipital_Mid_L | 51 | 6 | 6 | 0.12 | Frontal_Mid_L | 7 | 272 | 244 | 43.06 |
| Parietal_Inf_L | 61 | 8 | 6 | 0.15 | Supp_Motor_Area_L | 19 | 215 | 205 | 23.82 |
| Frontal_Sup_Orb_R | 6 | 6 | 5 | 0.08 | Parietal_Sup_L | 59 | 224 | 167 | 18.63 |
| Frontal_Mid_L | 7 | 11 | 5 | 0.15 | Temporal_Mid_R | 86 | 175 | 154 | 27.18 |
| Frontal_Mid_Orb_R | 10 | 5 | 5 | 0.08 | Frontal_Sup_L | 3 | 155 | 147 | 21.81 |

# 6 DISCUSSION AND CONCLUSION

In this work, we have proposed the ST-CAR prior, which is a general and flexible prior that could be applied to any regression problems with imaging components. Variational inference algorithms are proposed for the ST-CAR prior. Especially, we implemented the coordinate ascent variational inference (CAVI) as a baseline VI algorithm that can provide good estimation accuracy in low SNR settings, and we proposed a novel stochastic subsampling variational inference (SSVI) algorithm that is more computationally efficient. We demonstrated the use of the ST-CAR prior in both scalar-on-image and image-on-scalar regression models. Through comparisons in numeric studies, we find our proposed method has better performance in terms of estimation and computation, compared with existing methods such as T-LoHo, STGP, and SBIOS. The proposed method is applied to the ABCD study with task fMRI image data and identifies the left Precuneus as a significant region to the children's IQ development, and the development of the middle frontal gyrus as the significant region that can be most positively impacted by parental education level.

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

# A APPENDIX

## A.1 DETAILS ON IONS REGRESSION

For confounder coefficients $\xi_k$ and the individual effects $\eta_i$, we assign the Gaussian Process prior with the same kernel function $\kappa$ for computational convenience. The individual effect parameter $\eta_i$ separates the spatially correlated noise from the noise term $\epsilon_i$, and avoids setting a dense correlation matrix for the noise term $\epsilon_i$, which speeds up the computation. This is similar to the correlated noise model in Zhu et al. (2014). The identifiability of model equation 4 has been shown in Zhang et al. (2023a) under the following sufficient conditions: (1) the design matrix $\tilde{\mathbf{X}} := (\mathbf{X}, \mathbf{C}) \in \mathbb{R}^{n \times (m+1)}$ is a full rank matrix, (2) for any $i$ and any $s_j$, denote $\boldsymbol{\eta}(s_j) = (\eta_1(s_j), \ldots, \eta_n(s_j)) \in \mathbb{R}^n$, $\tilde{\mathbf{X}}^T \boldsymbol{\eta}(s_j) = 0$. The first condition is easily satisfied when the design matrix $\tilde{\mathbf{X}}$ is not linearly dependent.

For the Gaussian Process prior update of $\xi_k$ and $\eta_i$, we use the basis decomposition approach. Leveraging Mercer's theorem, which asserts that for any function $g(s)$ following a Gaussian Process with mean zero and covariance function $\sigma_g^2 \kappa(\cdot, \cdot)$, we can utilize the following basis decomposition.

$$g(s) = \sum_{l=1}^{\infty} \theta_{g,l} \phi_l(s), \quad \theta_{g,l} \stackrel{\text{ind}}{\sim} \mathcal{N}(0, \sigma_g^2 \lambda_l),$$

where $\lambda_l$ is the $l$-th eigen-value, and $\phi_l$ is the $l$-th eigen-function (see Section 4.2 in Rasmussen & Williams (2005)). In practice, we choose a finite $L$ as the cutoff on the number of bases, and approximate $g(s)$ by $\sum_{l=1}^{L} \theta_{g,l} \phi_l(s)$. The number of basis $L$ is chosen such that the summation $\sum_{l=1}^{L} \lambda_l$ is over 90% of $\sum_{l=1}^{p} \lambda_l$. The choice of the kernel function includes exponential square kernel, Matérn kernel, and other kernel functions. For the simulation section, we use the modified exponential square kernel, $\kappa(s, s'; a, b) = \exp\{-a(s^2 + s'^2) - b(s - s')^2\}$. For the real data analysis with ABCD data, the kernel is a pre-tuned Matérn kernel with region-specific smoothness parameters that can best align with the empirical correlation of the observed image data, same as in Xu & Kang (2023).

## A.2 COMPUTATIONAL DETAILS

For the neighborhood matrix $B$ in ST-CAR$(\nu, B)$, in order to speed up the computation, we use sparse matrix structure in RcppArmadillo (Eddelbuettel & Sanderson, 2014), and set a fixed bandwidth $|\mathcal{N}(j)|$ for all $j$. For a given fixed grid $\{s_1, \ldots, s_p\}$ in $\mathbb{R}^d$, we use RANN package (Arya et al., 2019) to efficiently search for the nearest neighbors in the high-dimensional setting.

### A.3 STOCHASTIC SUBSAMPLING VARIATIONAL INFERENCE (SSVI)

To make the variational inference method scalable for large data sets, we propose a stochastic subsampling version of CAVI, referred to as SSVI. The main computational bottleneck of CAVI is to update $\boldsymbol{\beta}$, which is a high-dimensional parameter, and the latent variable $\boldsymbol{\mu}$ further requires complex computation of mixed truncated normal densities. Hence given $\boldsymbol{\mu}$, when updating $\boldsymbol{\beta}$, we randomly select a subsample of data, indexed by $I \subset \{1, \ldots, n\}$, and apply a stochastic gradient update similar to the Stochastic Gradient Langevine Dynamics (SGLD) (Welling & Teh, 2011). Let $s_t$ be the step size at $t$-th iteration, $n$ be the total number of observations, $n_s$ be the subsample size, and $\pi$ be the prior density of $\beta_j$ at the $j$th voxel,

$$\mathbb{E}_{q^{(t)}}\{\beta_j\} \leftarrow \mathbb{E}_{q^{(t-1)}}\{\beta_j\} + s_t \left(\frac{n}{n_s}\nabla\mathbb{E}_{q^{(t-1)}}\log\sum_{i\in I}p(Y_i, \mathbf{M}_i, \mathbf{X}_i \mid \theta) + \nabla\mathbb{E}_{q^{(t-1)}}\log\pi(\beta_j)\right).$$

This is because under the mean-field assumption, the optimum density $q^*(\beta_j)$ has a closed-form solution: a normal density with mean and variance

$$\mathbb{E}_{q^*}(\beta_j) = \text{Var}_{q^*}(\beta_j)\times$$

$$\left[\mathbb{E}_{q^*}(\sigma_Y^{-2})\sum_{i=1}^{N}M_{i,j}\left(Y_i - \mathbb{E}_{q^*}\boldsymbol{\gamma}^T\mathbf{X}_i - \sum_{k\in[-j]}\mathbb{E}_{q^*}\beta_k M_{i,k} + \mathbb{E}_{q^*}\left\{\sigma_\beta^{-2}T_\nu(\mu_j)\right\}\right)\right]$$

$$\text{Var}_{q^*}(\beta_j) = \left(\mathbb{E}_{q^*}(\sigma_Y^{-2})\sum_{i=1}^{N}M_{i,j}^2 + \mathbb{E}_{q^*}(\sigma_\beta^{-2})\right)^{-1}$$

And $\mathbb{E}_{q^*}(\beta_j)$ is also the maximizer to

$$\mathbb{E}_{q^*}\sum_{i=1}^{N}\log p(Y_i, \mathbf{M}_i, \mathbf{X}_i \mid \theta) + \mathbb{E}_{q^*}\log\pi(\beta_j).$$

We require the step size $s_t$ to decrease to 0 as $t \rightarrow \infty$. In practice, we use the decay function $s_t = a(b + t)^{-\gamma}$, as suggested in Welling & Teh (2011).

In practice, we find that in low signal-to-noise ratio (SNR) settings, the CAVI algorithm gives better accuracy. Hence we recommend using CAVI for the SonI model, where the SNR can be very low, especially in brain imaging data, and using SSVI for the IonS model since the IonS model has a much higher SNR for the coefficient at each voxel. The implementation is available as an R package STCAR in the supplementary material[5].

### A.4 SIMULATION II: IMAGE-ON-SCALAR REGRESSION WITH SSVI

For IonS model equation 4, we compare ST-CAR with 3 other methods: (1) STGP prior, (2) Scalable Bayesian Image-on-Scalar regression (SBIOS) (Xu et al., 2024), (3) Mass Univariate Analysis (MUA). For the IonS regression equation 4, estimation of $\alpha$ has a larger SNR compared to estimating $\beta$ in SonI equation 3, hence we use SSVI for this application for ST-CAR prior. Because we impose GP prior for the confounder parameters $\xi_k$ and individual effect $\eta_i$, the GP kernels used in this simulation are all the same for STGP, ST-CAR, and SBIOS for fair comparison. We also use region-wise independent kernels for the GP priors in equation 4. The GP kernel is the same as equation 5 with $a = 0.01$ and $b = 10$.

The mass univariate analysis (MUA) is one of the most commonly used methods for IonS regression. MUA analyzes IonS as a spatially independent problem and treats the IonS regression as $p$ independent linear regression problems with exposure $X_i$ and confounders $\mathbf{C}_i$. To select active voxels, we use the Benjamini-Hochberg adjusted p-values (Benjamini & Hochberg, 1995) to control the false discovery rate. The active voxels selected by MUA have an adjusted p-value below 0.05.

The STGP method is similar to what has been discussed in the SonI regression. For the IonS regression, we use a total of $2 \times 10^4$ iterations and take the last 10% as the converged MCMC sample. The

---

[5]STCAR is also available on Github `https://github.com/yuliangxu/STCAR`

thresholding parameter $\nu$ is set to be 0.2. We use the point estimates of $\alpha$ and $\xi_k$ from MUA as the initial value for the MALA algorithm.

The Scalable Bayesian Image-on-Scalar regression (SBIOS) (Xu et al., 2024) is another Bayesian approach where the parameter of interest can be expressed as $\alpha(s) = \tilde{\alpha}(s)\delta(s)$. The latent spatially smooth function $\tilde{\alpha}$ is assigned a GP prior, and the binary selection variable $\delta(s_j)$ is assigned an independent prior $\text{Ber}(p(s_j))$ for each location $s_j$. SBIOS is designed to analyze a large-scale data set by using batch updates with stochastic gradient Langevin dynamics algorithm (SGLD). Hence it is more appropriate to be compared with the SSVI implementation of ST-CAR, since both methods are based on stochastic gradient updates of a small random sample drawn from the entire observed data. Different from the idea of SSVI where we simply use stochastic gradient update for an optimization problem, SGLD gives a smooth transition from optimization to MCMC sampling as the step size decays to 0 (Welling & Teh, 2011). Similar to STGP, we can use the MCMC sample of $\delta(s_j)$ to determine the PIP at location $j$, $\text{PIP}_j = \sum_{t=1}^{T} \delta(s_j)^{(t)} \neq 0/T$ for $T$ MCMC sample. In the simulation, we use 5000 SGLD iterations, with the decay function of the step size set as $s_t = 0.0001 \cdot (10 + t)^{-0.35}$. We use 200 subsample in each iteration. The prior for $\delta(s_j)$ is set to be $\text{Ber}(0.5)$ for all locations. The last 20% of iterations is used to compute the point estimation of $\alpha$ and PIP.

The ST-CAR method implemented using the SSVI algorithm requires a stochastic gradient update of $\alpha$. We use a step of $10^{-4}$ and a subsample of 100 for the SGD optimization. The decay rate function for $\sigma_\alpha^2$ is $(1 + t)^{-0.4}$. We use $C^+(1)$ as the prior for $\sigma_\mu$ in equation 1. Because of the randomness in the SGD update, we cannot use the difference between $\alpha^{(t)}$ and $\alpha^{(t+1)}$ or ELBO as a stopping rule. For the simulation, we simply run $10^4$ iterations. In practice, the convergence of SSVI can be roughly determined by the convergence of $\sigma_\mu^2$. For the point estimation and inference of $\alpha$, we use the averaged values over the last 20% iterations as the posterior mean of $\alpha$ and PIP to avoid the randomness from SGD.

Note that updating the individual effects $\eta_i$ for $i = 1, \ldots, n$ is computationally challenging for all Bayesian methods. We choose to update $\eta_i$ every 100 iterations for ST-CAR, SBIOS, and every 1000 iterations for STGP.

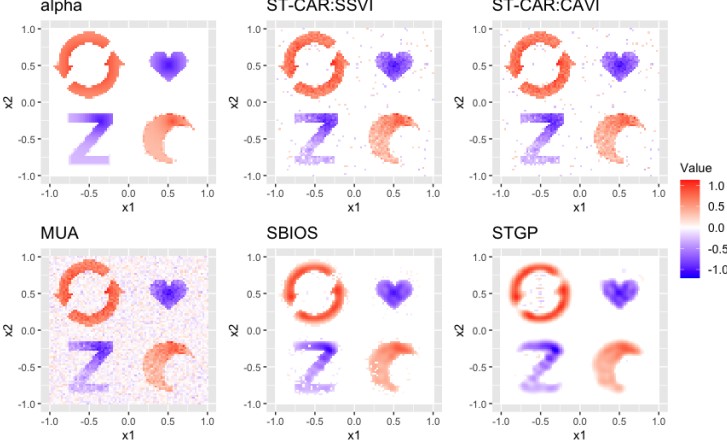

Figure A1: Point estimation result of IonS regression for all competing methods, $n = 600$, $p = 6400$, $\sigma_M^2 = 5$. The top-left figure is the true $\alpha$ signal.

Figure A1 provides a visualization of the point estimation for each method. MUA has the most noisy point estimation since it does not consider the spatial correlation, and there is no sparsity constraint directly imposed other than using the adjusted p-value to determine the level of significance for each voxel location. STGP suffers from the same issue as in Figure 2, where the pre-specified kernel is too smooth for the Z-shape and recycle shape (top-left). SBIOS uses the same kernel, but the binary selection parameter $\delta(s_j)$ has a spatially independent prior, and can get a clearer edge compared to STGP and better selection, but the latent GP kernel is still too smooth that the edge of the recycle shape and Z-shape tends to be 0. In the ST-CAR plot, although the functional parameters $\xi_k, \eta_i$

are all assigned GP prior with the same kernel as STGP and SBIOS, we can still see that ST-CAR can give a very clear edge for all 4 shapes. This demonstrates that ST-CAR prior is very flexible to different correlation patterns without much tuning on the neighborhood matrix $B$ or correlation coefficient $\rho$, especially for the high SNR cases.

Table A1: Numeric result for IonS simulation, under 100 replications.

(a) IonS: Comparison of estimation accuracy. The evaluation criteria include false discovery rate (FDR), true positive rate (TPR), overall accuracy (ACC), and root mean squared error (RMSE), all multiplied by 100.

| Case 1. $n = 600, p = 1600, \sigma_M^2 = 5$ | | | | | Case 2. $n = 600, p = 900, \sigma_M^2 = 5$ | | | |
|---|---|---|---|---|---|---|---|---|
| Criteria | ST-CAR | MUA | STGP | SBIOS | Criteria | ST-CAR | MUA | STGP | SBIOS |
| FDR | 5.8 | 7.98 | 16.88 | 6.3 | FDR | 4.95 | 8.23 | 12.75 | 4.69 |
| TPR | 95.41 | 93.2 | 94.18 | 94.94 | TPR | 84.34 | 81.42 | 85.01 | 84.61 |
| ACC | 97.86 | 96.95 | 94.89 | 97.66 | ACC | 96.27 | 95.19 | 94.9 | 96.36 |
| RMSE | **7.86** | 9.35 | 10.53 | 10.79 | RMSE | 7.88 | 9.4 | **6.85** | 7.36 |
| Case 3. $n = 1000, p = 1600, \sigma_M^2 = 5$ | | | | | Case 4. $n = 600, p = 1600, \sigma_M^2 = 10$ | | | |
| Criteria | ST-CAR | MUA | STGP | SBIOS | Criteria | ST-CAR | MUA | STGP | SBIOS |
| FDR | 7.1 | 8.1 | 19.24 | 7.97 | FDR | 5.07 | 8.06 | 14.87 | 4.12 |
| TPR | 98.38 | 97.32 | 95.57 | 97.53 | TPR | 85.31 | 82.61 | 90.84 | 86.19 |
| ACC | 98.13 | 97.69 | 94.43 | 97.76 | ACC | 96.06 | 94.96 | 94.88 | 96.42 |
| RMSE | **6.44** | 7.21 | 10.16 | 10.35 | RMSE | **10.32** | 13.14 | 11.21 | 11.74 |
| Case 5. $n = 600, p = 6400, \sigma_M^2 = 5$ | | | | | Case 6. $n = 1000, p = 6400, \sigma_M^2 = 20$ | | | |
| Criteria | ST-CAR | MUA | STGP | SBIOS | Criteria | ST-CAR | MUA | STGP | SBIOS |
| FDR | 5.97 | 7.84 | 20.78 | 6.04 | FDR | 5.93 | 7.93 | 19.94 | 2.95 |
| TPR | 93.64 | 91.78 | 97.19 | 93.62 | TPR | 81.83 | 80.09 | 96.64 | 84.46 |
| ACC | 97.35 | 96.55 | 93.9 | 97.33 | ACC | 95.06 | 94.32 | 94.18 | 96.16 |
| RMSE | **8.52** | 9.19 | 9.79 | 10.25 | RMSE | 13.84 | 14.19 | **9.76** | 10.72 |

(b) Computation time for IonS simulation, averaged over 100 replications.

| Computation time | Total time (seconds) | | | Number of iterations per second | | |
|---|---|---|---|---|---|---|
| Case | ST-CAR | STGP | SBIOS | ST-CAR | STGP | SBIOS |
| Case 1. $n = 600, p = 1600, \sigma_M^2 = 5$ | 73.7 | 588.4 | 717.4 | 137.5 | 3.4 | 7.3 |
| Case 2. $n = 600, \boldsymbol{p = 900}, \sigma_M^2 = 5$ | 55 | 381.6 | 874.5 | 186 | 5.3 | 6.3 |
| Case 3. $\boldsymbol{n = 1000}, p = 1600, \sigma_M^2 = 5$ | 117.3 | 1062.1 | 1968.4 | 88.9 | 1.9 | 3.1 |
| Case 4. $n = 600, p = 1600, \boldsymbol{\sigma_M^2 = 10}$ | 82.9 | 621.8 | 1214.1 | 122.7 | 3.2 | 4.9 |
| Case 5. $n = 600, p = 6400, \boldsymbol{\sigma_M^2 = 5}$ | 409.2 | 2190.3 | 1049.8 | 24.7 | 0.9 | 5.3 |
| Case 6. $n = 600, p = 6400, \boldsymbol{\sigma_M^2 = 20}$ | 596.7 | 5090.2 | 1733.2 | 17 | 0.4 | 3.2 |

Table A1a provides the numerical result on the IonS model based on 100 replications in six different settings. Because the predictive MSE on the outcome averaged over all voxel locations is very close for all methods, we do not report it here. Instead, we focus on the estimation of the coefficient $\alpha$. The proposed ST-CAR prior with the SSVI algorithm gives the lowest RMSE except for Case 2 and 6, for which the STGP has the lowest RMSE, although STGP has a much larger FDR in both cases. To control the FDR below 10%, we use the Benjamini-Hochberg (BH) adjusted p-values on MUA and set a threshold such that $\alpha(s_j)$ with the adjusted p-value below 0.1 are selected as active voxels. For ST-CAR, STGP, and SBIOS, we compute the proportion of the active voxels selected by MUA, and apply the same proportion to get the cutoff on PIP. In this way, we select roughly the same proportion of voxels as active. Based on the result in Table A1a, this selection method can control the FDR for ST-CAR and SBIOS to be below 10%, whereas for STGP, the FDR is still over 10%. The MUA has the worst power (TPR) in all scenarios after controlling for FDR. The total running time shown in Table A1b also shows a great improvement in the computational speed for the SSVI algorithm when compared with other MCMC sampling algorithms. On average, STGP takes 7.6 times longer compared to SSVI, and SBIOS takes 10.4 times longer compared to SSVI. In the Supplemental Material, we also provide additional results of using CAVI under ST-CAR prior and compare the performance with SSVI. SSVI still slightly outperforms CAVI in the IonS regression in terms of both estimation and computation speed.

For a more comprehensive comparison between different estimation algorithms (Gibbs, CAVI, SSVI) under ST-CAR prior, we include a small low-dimensional comparison for the SonI regression in the Supplemental Material. The result suggests that CAVI tends to have better estimation accuracy in SonI regression where the signal-to-noise ratio is low.

## A.5    POSTERIOR DERIVATION

For the fully conjugate posterior derivations, the hierarchical model of applying the sparse-mean prior on the scalar-on-image regression can be written as

$$Y_i = \sum_{j=1}^p \beta(s_j) M_i(s_j) + \boldsymbol{\gamma}^{\mathrm{T}} \mathbf{X}_i + \epsilon_i \quad \epsilon_i \overset{\text{iid}}{\sim} \boldsymbol{N}(0, \sigma_Y^2), \quad \sigma_Y \sim C^+(1)$$

$$\beta(s_j) \overset{\text{ind}}{\sim} \boldsymbol{N}(T_\nu(\mu_j), \sigma_\beta^2), \quad \sigma_\beta \sim C^+(1)$$

$$\mu_j \mid \mu_{[-j]} \sim \boldsymbol{N}(\bar{\mu}_{\mathcal{N}(j)}, \tau_j^2), \quad \tau_{\mu,j}^2 = \sigma_\mu^2 / w_{j+}, \quad \sigma_\mu^2 \sim C^+(1)$$

$$\boldsymbol{\gamma} \sim \boldsymbol{N}(0, \sigma_\gamma^2 I_q), \quad \sigma_\gamma \sim C^+(1)$$

Based on this hierarchical model, we can derive the posterior distributions of each parameter. Note that the posterior for most of the parameters is straightforward, except for $\mu_j$, which involves the soft-thresholding operator $T_\nu$. The posterior of $\mu_j$ can be expressed in terms of a mixture of truncated normal distributions with 3 components.

### A.5.1    PROOF OF PROPOSITION 1

*Proof.* The posterior of $\mu_j$ can be expressed as

$$\log \pi(\mu_j \mid \boldsymbol{\beta}, \mu_{[-j]}, \sigma_\mu, \sigma_\beta) \propto -\frac{1}{2\sigma_\beta^2} (\beta(s_j) - T_\nu(\mu_j))^2 - \frac{w_{j+}}{2\sigma_\mu^2} (\mu_j - \bar{\mu}_{\mathcal{N}(j)})^2 \tag{6}$$

$$\pi(\mu_j \mid \dots) = P_j^+ \cdot \boldsymbol{N}_{[\nu,+\infty)}(\mu_j^+, V_j) + P_j^0 \cdot \boldsymbol{N}_{[-\nu,\nu]}(\bar{\mu}_{\mathcal{N}(j)}, V_0) + P_j^- \cdot \boldsymbol{N}_{(-\infty,-\nu]}(\mu_j^-, V_j)$$

where $\boldsymbol{N}_{[a,b]}(\mu, \sigma^2)$ is notation for the truncated normal distribution supported on $[a, b]$ with mean $\mu$ and variance $\sigma^2$. The middle component is just the truncated normal on $[-\nu, \nu]$ with the prior mean $\bar{\mu}_{\mathcal{N}(j)}$ and variance $V_0 = \frac{w_{j+}}{\sigma_\mu^2}$. For the other two components,

$$V_j = \left( \frac{1}{\sigma_\beta^2} + \frac{w_{j+}}{\sigma_\mu^2} \right)^{-1}, \quad \mu_j^+ = V_j \left\{ \frac{1}{\sigma_\beta^2} (\beta(s_j) + \nu) + \frac{w_{j+}}{\sigma_\mu^2} \bar{\mu}_{\mathcal{N}(j)} \right\},$$

$$\mu_j^- = V_j \left\{ \frac{1}{\sigma_\beta^2} (\beta(s_j) - \nu) + \frac{w_{j+}}{\sigma_\mu^2} \bar{\mu}_{\mathcal{N}(j)} \right\}$$

The density of this 3 component mixture can be expressed as

$$\pi(\mu_j \mid \dots) = \frac{1}{Z_j} (Z_j^+ f_j^+ + Z_j^0 f_j^0 + Z_j^- f_j^-)$$

where $Z_j^+, Z_j^0, Z_j^-, Z_j$ represent different normalizing constant, and $f_j^+, f_j^0, f_j^-$ represent the density functions of 3 truncated normal distributions $\boldsymbol{N}_{[\nu,+\infty)}(\mu_j^+, V_j), \boldsymbol{N}_{[-\nu,\nu]}(\bar{\mu}_{\mathcal{N}(j)}, V_0), \boldsymbol{N}_{(-\infty,-\nu]}(\mu_j^-, V_j)$ respectively. Hence the mixing probabilities can be represented as

$$P_j^+ = \frac{Z_j^+}{Z_j}, \quad P_j^0 = \frac{Z_j^0}{Z_j}, \quad P_j^- = \frac{Z_j^-}{Z_j}.$$

Now denote $\tilde{f}_*, * \in \{-, 0, +\}$ as the RHS in equation 6 supported on $x \in (-\infty, -\nu), [-\nu, \nu], (\nu, +\infty)$ respectively.

$$\log(Z_j^+) = \log \tilde{f}_j^+ - \log f_j^+, \quad x \in (\nu, +\infty)$$

$$= -\frac{1}{2\sigma_\beta^2} (\beta(s_j) + \nu - \mu_j)^2 - \frac{w_{j+}}{2\sigma_\mu^2} (\mu_j - \bar{\mu}_{\mathcal{N}(j)})^2$$

$$- \left\{ \log \frac{1}{\sqrt{V_j}} - \frac{1}{2V_j} (\mu_j - \mu_j^+)^2 - \log \left( 1 - \Phi \left( \frac{\nu - \mu_j^+}{\sqrt{V_j}} \right) \right) \right\}$$

$$\log(Z_j^0) = \log \tilde{f}_j^0 - \log f_j^0, \quad x \in [-\nu, +\nu]$$

$$= -\frac{1}{2\sigma_\beta^2} \left(\beta(s_j)\right)^2 - \frac{w_{j+}}{2\sigma_\mu^2} \left(\mu_j - \bar{\mu}_{\mathcal{N}(j)}\right)^2$$

$$- \left\{ \log \frac{1}{\sqrt{V_0}} - \frac{1}{2V_0} \left(\mu_j - \bar{\mu}_{\mathcal{N}(j)}\right)^2 - \log \left(1 - \Phi\left(\frac{\nu - \bar{\mu}_{\mathcal{N}(j)}}{\sqrt{V_0}}\right)\right) \right\}$$

$$\log(Z_j^-) = \log \tilde{f}_j^- - \log f_j^-, \quad x \in (-\infty, -\nu)$$

$$= -\frac{1}{2\sigma_\beta^2} \left(\beta(s_j) - \mu_j - \nu\right)^2 - \frac{w_{j+}}{2\sigma_\mu^2} \left(\mu_j - \bar{\mu}_{\mathcal{N}(j)}\right)^2$$

$$- \left\{ \log \frac{1}{\sqrt{V_j}} - \frac{1}{2V_j} \left(\mu_j - \mu_j^-\right)^2 - \log \left(1 - \Phi\left(\frac{\nu - \mu_j^-}{\sqrt{V_j}}\right)\right) \right\}$$

Hence the entire density function is complete. $\qquad\square$

### A.5.2 Variational inference: Q-densities for scalar-on-image regression

In the following derivation, we denote the vector $Y \in \mathbb{R}^n$, matrix $M \in R^{n \times p}$, $X \in R^{n \times q}$ to denote the outcome and design matrices.

**Q-density for $\beta$ using SVD**

First, we use Singular Value Decomposition (SVD) on $M$ and re-express the scalar-on-image regression model as follows.

Let the compact SVD of $M \in \mathbb{R}^{n \times p}$ be $M = UDV^T$ where $U \in \mathbb{R}^{n \times n}, V^T \in \mathbb{R}^{n \times p}$, and $U^T U = UU^T = I_n, V^T V = I_n$. Let $\tilde{\beta} = \beta - T_\nu(\mu)$, $\tilde{Y} = Y - MT_\nu(\mu) - X\gamma = M\tilde{\beta} + \epsilon$.

Now apply the rotation matrix $U$ on both sides, $\tilde{Y}^* = U^T \tilde{Y} = DV^T \beta + \epsilon$. The q-density for $\tilde{\beta}$ is now a normal density with mean and variance

$$\text{Var}_q(\tilde{\beta}|\sim) = \left(\mathbb{E}_q\left(\frac{1}{\sigma_\beta^2}\right) I_p + \mathbb{E}_q\left(\frac{1}{\sigma_Y^2}\right) VD^T DV^T\right)^{-1},$$

$$\mathbb{E}_q(\tilde{\beta}|\sim) = \text{Var}_q(\tilde{\beta}|\sim) \left(\mathbb{E}_q\left(\frac{1}{\sigma_Y^2}\right) VD^T \tilde{Y}^*\right).$$

Note that $\mathbb{E}_q(\tilde{\beta}|\sim)$ can be further simplified,

$$\mathbb{E}_q(\tilde{\beta}|\sim) = VD\left(\frac{1}{\tau^2} I_n + D^2\right)^{-1} \tilde{Y}^*$$

where $\tau^2 = \dfrac{\mathbb{E}_q\left(\frac{1}{\sigma_Y^2}\right)}{\mathbb{E}_q\left(\frac{1}{\sigma_\beta^2}\right)}$. Then $\mathbb{E}_q(\beta|\sim) = \mathbb{E}_q(\tilde{\beta}|\sim) + T_\nu(\mu)$.

The q-density for $\sigma_\beta$ is as follows. Note that we use the hierarchical expression to sample half-Cauchy prior $\sigma_\beta \sim C^+(1)$: $\sigma_\beta^2 \sim IG(\frac{1}{2}, \frac{1}{a_\beta})$, $a_\beta \sim IG(\frac{1}{2}, 1)$.

$$\log \pi(\sigma_\beta^2|\sim) \propto -\frac{1}{2\sigma_\beta^2} \sum_{j=1}^{p} \left(\beta(s_j) - T_\nu(\mu_j)\right)^2 - \frac{p}{2}\log(\sigma_\beta^2) - \left(\frac{1}{2} + 1\right)\log\sigma_\beta^2 - \frac{1}{a_\beta}\frac{1}{\sigma_\beta^2}$$

Hence the q-density for $\sigma_\beta^2$ follows $IG\left(\frac{p+1}{2}, \frac{1}{2}\sum_{j=1}^{p} \mathbb{E}_q\left(\beta(s_j) - T_\nu(\mu_j)\right)^2 + \frac{1}{a_\beta}\right)$. Note that $\mathbb{E}_q\|\beta - T_\nu(\mu)\|_2^2 = \text{Tr}(\text{Var}(\beta)) + \|\mathbb{E}_q(\beta) - T_\nu(\mu)\|_2^2$, and the marginal variance $\text{Var}_q(\beta(s_j)) = \left[\mathbb{E}_q\left(\frac{1}{\sigma_Y^2}\right)\sum_{i=1}^{n} M_i(s_j)^2 + \mathbb{E}_q\left(\frac{1}{\sigma_\beta^2}\right)\right]^{-1}$.

The q-density for $a_\beta$ is $IG\left(1, 1 + \mathbb{E}\left\{\frac{1}{\sigma_\beta^2}\right\}\right)$.

**Q-density for** $\gamma$ The q-density of $\gamma$ follows the multivariate normal distribution with mean and variance

$$\text{Var}_q(\gamma|\sim) = \left\{ \mathbb{E}_q\left(\frac{1}{\sigma_Y^2}\right) X^T X + \mathbb{E}_q\left(\frac{1}{\sigma_\gamma^2} I_q\right) \right\}^{-1}$$

$$\mathbb{E}(\gamma|\sim) = \text{Var}_q(\gamma|\sim) \left\{ \mathbb{E}_q\left(\frac{1}{\sigma_Y^2}\right) \sum_i \left(Y_i - M_i^T \beta\right) X_i \right\}$$

To speed up the computation, we use eigen-decomposition $X^T X = Q\Lambda_X Q^T$, and the variance update can be written as

$$\text{Var}_q(\gamma|\sim) = Q\text{diag}\left\{ \mathbb{E}_q\left(\frac{1}{\sigma_Y^2}\right) \Lambda_X + \mathbb{E}_q\left(\frac{1}{\sigma_\gamma^2}\right) \right\}^{-1} Q^T.$$

Similarly, the q-density for $\sigma_\gamma^2$ follows IG $\left(\frac{q+1}{2}, \frac{1}{2}\mathbb{E}_q\|\gamma\|_2^2 + \mathbb{E}_q\left(\frac{1}{a_\gamma}\right)\right)$.

The q-density for $a_\gamma$ is IG $\left(1, 1 + \mathbb{E}\left\{\frac{1}{\sigma_\gamma^2}\right\}\right)$.

**Q-density for** $\sigma_Y$

$$\sigma_Y^2 \overset{q}{\sim} \text{IG}\left(\frac{n+1}{2}, \frac{1}{2}\mathbb{E}_q\|Y - M\beta - X\gamma\|_2 + \mathbb{E}_q\left(\frac{1}{a_Y}\right)\right)$$

$$a_Y \overset{q}{\sim} \text{IG}\left(1, 1 + \mathbb{E}_q\left(\frac{1}{\sigma_Y^2}\right)\right)$$

**ELBO derivation**

$$\begin{aligned}
\text{ELBO} =& \mathbb{E}_q\left\{\log \pi(Y \mid M, X, \beta, \gamma, \sigma_\beta^2, \sigma_\gamma^2, \sigma_Y^2)\right\} \\
& - \mathbb{E}_q\left\{\log q(\beta) + \log q(\gamma) + \log q(\sigma_\beta^2) + \log q(\sigma_\gamma^2) + \log q(\sigma_Y^2)\right\} \\
=& \mathbb{E}_q\left\{\log \pi(Y|\sim)\right\} \\
& + \mathbb{E}_q\left\{\log \pi(\beta|\sim) - \log q(\beta)\right\} + \mathbb{E}_q\left\{\log \pi(\mu)|\sim) - \log q(\pi(\mu))\right\} \\
& + \mathbb{E}_q\left\{\log \pi(\sigma_\beta^2|\sim) - \log q(\sigma_\beta^2)\right\} + \mathbb{E}_q\left\{\log \pi(a_\beta|\sim) - \log q(a_\beta)\right\} \\
& + \mathbb{E}_q\left\{\log \pi(\gamma|\sim) - \log q(\gamma)\right\} + \mathbb{E}_q\left\{\log \pi(\sigma_\gamma^2|\sim) - \log q(\sigma_\gamma^2)\right\} + \mathbb{E}_q\left\{\log \pi(a_\gamma|\sim) - \log q(a_\gamma)\right\} \\
& \mathbb{E}_q\left\{\log \pi(\sigma_Y^2|\sim) - \log q(\sigma_Y^2)\right\} + \mathbb{E}_q\left\{\log \pi(a_Y|\sim) - \log q(a_Y)\right\}
\end{aligned}$$

In the implementation, we separately compute each part of the ELBO and add them together.

$$\begin{aligned}
\text{ELBO}_{\text{logL}} =& \mathbb{E}_q\left\{\log \pi(Y \mid M, X, \beta, \gamma, \sigma_\beta^2, \sigma_\gamma^2, \sigma_Y^2)\right\} \\
=& \frac{n}{2}\mathbb{E}_q\left(\frac{1}{\sigma_Y^2}\right) - \frac{1}{2}\mathbb{E}_q\left(\frac{1}{\sigma_Y^2}\right)\mathbb{E}_q\|Y - M\beta - X\gamma\|_2^2
\end{aligned}$$

Here, denote $\mathbb{E}_q\text{SSE} = \mathbb{E}_q\|Y - M\beta - X\gamma\|_2^2$,

$$\mathbb{E}_q\text{SSE} = \|Y - M\mathbb{E}_q\beta - X\mathbb{E}_q\gamma\|_2^2 + \text{Tr}\left\{M^T M\text{Var}_q(\beta)\right\} + \text{Tr}\left\{X^T X\text{Var}_q(\gamma)\right\}.$$

With the eigen decomposition on $X^T X$,

$$\text{Tr}\left\{X^T X\text{Var}_q(\gamma)\right\} = \text{Tr}\left\{\Lambda_X \text{diag}\left\{\mathbb{E}_q\left(\frac{1}{\sigma_Y^2}\right)\Lambda_X + \mathbb{E}_q\left(\frac{1}{\sigma_\gamma^2}\right)\right\}^{-1}\right\}$$

## A.6 SENSITIVITY ANALYSIS TO CHOOSE HYPERPARAMETERS

In addition, Table A3 provides additional sensitivity analysis results on SonI when the bandwidth is 26. Table A4 provides additional sensitivity analysis results on IonS when the bandwidth is 9 and the decay rate parameter $\gamma$ is 0.35.

Table A2: Sensitivity Analysis on SonI and IonS regressions.

(a) SonI for varying $\nu$ and initial value for $\sigma_\beta^2$. The Elastic Net and STGP results are shown as a comparison. Bandwidth=9 in the ST-CAR model. Additional sensitivity analysis where bandwidth=26 and varying decay rate $\gamma$ for $\sigma_\beta^2$ is available in the Appendix.

| $\sigma_\beta^2$ | $10^{-5}$ | $10^{-5}$ | $10^{-5}$ | $10^{-5}$ | $5\times10^{-5}$ | $10^{-5}$ | $5\times10^{-6}$ | ElasNet | STGP |
|---|---|---|---|---|---|---|---|---|---|
| $\nu$ | 0.003 | 0.005 | 0.007 | 0.01 | 0.005 | 0.005 | 0.005 | | |
| test pMSE | 0.58 | 0.5 | **0.48** | 0.48 | 0.61 | 0.5 | 0.55 | 0.53 | 0.5 |
| Test $R^2$ | 0.16 | 0.28 | **0.30** | 0.30 | 0.12 | 0.28 | 0.20 | 0.23 | 0.28 |
| train pMSE | 0.17 | 0.24 | 0.27 | 0.29 | 0.11 | 0.24 | 0.21 | 0.45 | 0.49 |
| Train $R^2$ | 0.75 | 0.65 | 0.61 | 0.58 | 0.84 | 0.65 | 0.70 | 0.35 | 0.29 |

(b) IonS for varying $\nu$, initial value for $\sigma_\alpha^2$, and decay rate $\gamma$ for $\sigma_\alpha^2$. The total test pMSE is the summation of all voxel-level pMSE. Bandwidth is 26. Additional sensitivity results where bandwidth=9 are available in the appendix.

| initial $\sigma_\alpha^2$ | 1 | 0.1 | 0.01 | 0.1 | 0.1 | 0.1 | 0.1 | 0.1 | 0.1 | |
|---|---|---|---|---|---|---|---|---|---|---|
| $\nu$ | 0.005 | 0.005 | 0.005 | 0.001 | 0.01 | 0.05 | 0.005 | 0.005 | 0.005 | MUA |
| Decay rate $\gamma$ | 0.35 | 0.35 | 0.35 | 0.35 | 0.35 | 0.35 | 0.25 | 0.45 | 0.55 | |
| total test pMSE | 47357.74 | 47351.78 | 47354.06 | 47354.13 | 47351 | 47354.1 | 47354.12 | **47350.7** | 47352.68 | 47487.05 |

Table A3: Additional sensitivity analysis for SonI when the bandwidth is 26, on three parameters: (i) the initial value of $\sigma_\beta^2$, (ii) the thresholding parameter $\nu$ in ST-CAR prior, (iii) the decay rate $\gamma$ in the decay rate function for $\sigma_\beta^2$ where $(\sigma_\beta^2)^{(t)} = a(b+t)^{-\gamma}$.

| $\sigma_\beta^2$ | $5\times10^{-6}$ | $1\times10^{-5}$ | $5\times10^{-5}$ | $1\times10^{-4}$ | $1\times10^{-5}$ | $1\times10^{-5}$ | $1\times10^{-5}$ | $1\times10^{-5}$ | $1\times10^{-5}$ |
|---|---|---|---|---|---|---|---|---|---|
| $\nu$ | 0.007 | 0.007 | 0.007 | 0.007 | 0.005 | 0.01 | 0.012 | 0.007 | 0.007 |
| $\gamma$ | 0.35 | 0.35 | 0.35 | 0.35 | 0.35 | 0.35 | 0.35 | 0.25 | 0.45 |
| test pMSE | 0.54 | 0.51 | 0.56 | 0.64 | 0.52 | 0.52 | 0.51 | 0.52 | 0.51 |
| train pMSE | 0.29 | 0.37 | 0.31 | 0.22 | 0.33 | 0.4 | 0.41 | 0.36 | 0.37 |

## A.7 ABCD DATA ANALYSIS USING ELASNET AND MUA

In this section, we perform the baseline methods ElasNet for SonI regression and MUA for IonS regression as a sanity check on the ABCD data analysis.

For SonI using Elastic Net with $\alpha = 0.5$ and $\lambda$ chosen with cross-validation, we visualize the point estimation on $\beta$ in Figure A2a. The Elastic Net has no straight forward uncertainty quantification method such as p-values, hence we can only compare the point estimates. As expected, the scale of $\beta$ using Elastic Net is also very small, and there are only a few small areas with $\beta$ near or greater than 0.001, as highlighted by the red areas.

For IonS using MUA, we note that after using BH-adjusted p-values in the MUA, none of the p-values can pass below the 0.05 threshold, which means MUA cannot identify any signal with a reasonable adjusted p-value. Nonetheless, we still visualize the point estimation of $\alpha$ on the whole brain. Based on Figure A2b, we can see that the MUA estimates of $\alpha$ with large positive effects pick up lots of noises, and compared with A1, the ST-CAR's estimate of active $\alpha$ is a subarea of the MUA's estimate and is validated by high PIP ($> 0.95$).

In addition, we present the $\beta$ estimation by ST-CAR after thresholding by $PIP > 0.25$ in Figure A3. In fact, both Figure A2a and Figure A3 demonstrate the true signals are very sparse in the SonI problem. ST-CAR still tends to identify more active voxels than ElasNet, and some active areas still overlap on the sagittal plane (first plot from the right) with ElasNet result.

To address the concern that the identified voxels in SonI might be due to random noises, we have conducted a stability check: we rerun the SonI on a slightly reduced subset of the ABCD data, by randomly removing 10% (or 30%) of the data to re-train the model, and check if the identified activation voxels in the full sample can still be recovered in each random subsample. We have conducted a total of 50 splits, and report the results in Table A5, where we combine the identified voxels by ST-CAR and ElasticNet in the full sample analysis to serve as the "ground truth". The numbers in each

Table A4: Additional sensitivity analysis for IonS when the bandwidth is 9, $\gamma = 0.35$ in the decay rate function, on two parameters: (i) the initial value of $\sigma_\beta^2$, (ii) the thresholding parameter $\nu$ in ST-CAR prior

| $\sigma_\alpha^2$ | 1 | 0.1 | 0.01 | 0.1 | 0.1 |
|---|---|---|---|---|---|
| $\lambda$ | 0.01 | 0.01 | 0.01 | 0.005 | 0.05 |
| total test pMSE | 47362.8 | 47351.85 | 47356.9 | 47354.97 | 47353.58 |

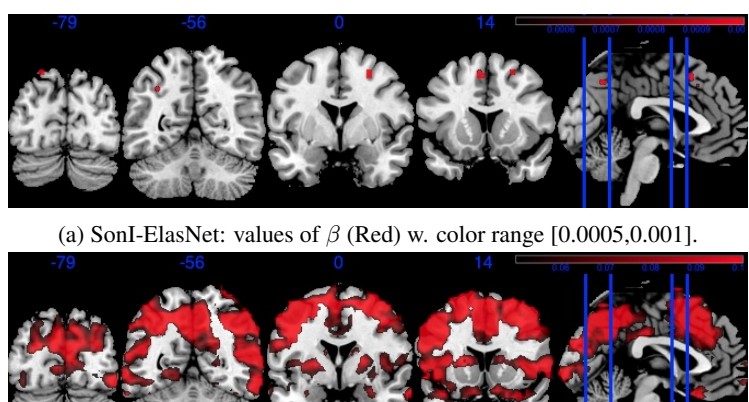

(a) SonI-ElasNet: values of $\beta$ (Red) w. color range [0.0005,0.001].

(b) IonS-MUA: point estimates of $\alpha$ (Red) w. color range [0.05,0.1].

Figure A2: Visual illustration of $\beta$ in SonI and $\alpha$ using MUA in IonS.

row represent the proportion of voxels being repeatedly selected in 50 splits. For example, in the top half of Table A5, 0.89 for row "ST-CAR: PIP>0.25" means that 89% of the voxels are selected by at least 10% of the 50 subsampled studies, with the activation criteria "PIP>0.25", similarly, 0.21 means 21% of voxels are repeatedly selected by at least 70% of the 50 subsampled studies.

Table A5: Result of stability check on SonI

| When randomly removing 10% of the data | | | | | | | | |
|---|---|---|---|---|---|---|---|---|
| threshold | 0.1 | 0.2 | 0.3 | 0.4 | 0.5 | 0.6 | 0.7 | 0.8 |
| ST-CAR: PIP>0.25 | 0.89 | 0.83 | 0.72 | 0.59 | 0.42 | 0.32 | 0.21 | 0.13 |
| ST-CAR: $\|\beta(s)\| > 5 \times 10^{-4}$ | 0.89 | 0.83 | 0.72 | 0.59 | 0.42 | 0.32 | 0.21 | 0.13 |
| ElasticNet: $\|\beta(s)\| > 5 \times 10^{-4}$ | 0.23 | 0.21 | 0.18 | 0.15 | 0.14 | 0.12 | 0.1 | 0.08 |
| When randomly removing 30% of the data: | | | | | | | | |
| threshold | 0.1 | 0.2 | 0.3 | 0.4 | 0.5 | 0.6 | 0.7 | 0.8 |
| ST-CAR: PIP>0.25 | 0.99 | 0.83 | 0.63 | 0.46 | 0.31 | 0.12 | 0.08 | 0.06 |
| ST-CAR: $\|\beta(s)\| > 5 \times 10^{-4}$ | 0.77 | 0.58 | 0.39 | 0.27 | 0.13 | 0.07 | 0.05 | 0.05 |
| ElasticNet: $\|\beta(s)\| > 5 \times 10^{-4}$ | 0.38 | 0.32 | 0.24 | 0.18 | 0.14 | 0.09 | 0.04 | 0.03 |

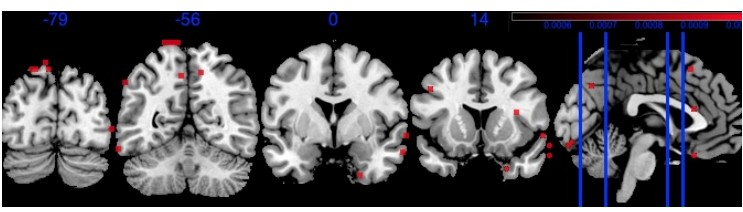

Figure A3: SonI: values of $\beta * I(PIP > 0.25)$ (Red) w. color range [0.0005,0.001].

The results show that, in both cases, ST-CAR consistently identifies more voxels than ElasticNet. This indicates that ST-CAR effectively borrows information across spatial locations, reducing susceptibility from noises compared to ElasticNet. The stability results indicate more robust performance of ST-CAR, particularly in scenarios with low SNR and unknown ground truth. This stability check approach has been widely applied in other neuroimaging studies where the underlying ground truth is not available and performance evaluation is challenging (Zhang et al., 2023b; Morris et al., 2022; Zhu et al., 2023).

### A.8 SCIENTIFIC EVIDENCE FOR THE MOTIVATION FOR ABCD STUDY

For the two scientific problems mentioned in Section 5, we provide more scientific evidence on the importance of this SonI and IonS problem in this section.

Recent studies (Cermakova et al., 2023; Halabicky et al., 2023) have demonstrated that parental education levels are significantly associated with children's cognitive abilities, including specific cognitive functions such as working memory. In particular, children's cognitive functions are often associated with their brain development, which fMRI captures during tasks like the working memory task in the ABCD study.

We chose to use general cognitive ability as the outcome because it provides a comprehensive assessment of a child's overall cognitive function, which encompasses not only working memory but also other critical skills such as reading, spelling, and math abilities (Alloway & Alloway, 2008). These abilities are often correlated and influenced by common neural processes, making general cognitive ability a robust and representative measure for examining broader cognitive development. By using this summary outcome, we can account for the cumulative effect of parental education on multiple facets of cognitive performance, providing a more holistic view of the relationship between socioeconomic factors and brain development.

### A.9 TABLES OF STANDARD DEVIATION FOR TABLE 1 AND A1A

In this section, we present the tables of standard deviations over 100 replicates for the mean values presented in Table 1 and A1a respectively, shown in Table A6.

### A.10 ADDITIONAL SIMULATION RESULTS

In the first simulation A1, we provide a low dimensional comparison on the SonI regression between three different implementation of the ST-CAR model, the Gibbs sampler, the CAVI algorithm, and the SSVI algorithm.

In the second simulation A2, we provide the additional to results to the simulation II IonS with a further comparison between CAVI and SSVI in high-dimensional settings.

### A.10.1 SIMULATION A1: LOW DIMENSIONAL COMPARISON (SONI)

We compare the proposed `Gibbs`, `CAVI`, `SSVI` with ST-CAR prior to the classical penalized regression method `glmnet`, and a Bayesian method where $\beta$ is assigned a Soft-thresholded Gaussian Process prior (Kang et al., 2018) implemented in the `BIMA` package.

The Frequentist penalized regression is implemented using R package `glmnet`(Friedman et al., 2010b) with lasso penalty ($\alpha = 1$), using 10-fold cross-validation.

The `BIMA` method requires a pre-specified kernel function, and the posterior sampling algorithm is Metropolis-adjusted Langevin algorithm (MALA). In this simulation we sue the exponential square kernel

$$\kappa(s, s'; a, b) = \text{cor}\{\beta(s), \beta(s')\} = \exp\{-a(s^2 + s'^2) - b(s - s')^2\}$$

where $a = 0.01, b = 10$, and used $L = 66$ basis functions.

For the four Bayesian methods (`Gibbs`, `CAVI`, `SSVI`, `BIMA`), we set the thresholding parameter $\nu = 0.1$. To evaluate the variable selection accuracy, for the variational inference ST-CAR methods (`CAVI`, `SSVI`), we use the mixing probabilities shown in 2 to define the posterior inclusion

Table A6: **Standard deviation** result for SonI simulation, under 100 replications.

(a) SonI: standard deviations over 100 replicates. The evaluation criteria for estimation includes false discovery rate (FDR), true positive rate (TPR), overall accuracy (ACC), and root mean squared error (RMSE), all multiplied by 100. The evaluation criteria for predictive performance includes training and testing predictive MSE, denoted as Train and Test pMSE respectively

| Case 1. $n=600, p=1600, \sigma_Y^2=1$ | | | | | Case 2. $n=600, \boldsymbol{p=900}, \sigma_Y^2=1$ | | | | |
|---|---|---|---|---|---|---|---|---|---|
| | ST-CAR | ElasNet | STGP | T-LoHo | | ST-CAR | ElasNet | STGP | T-LoHo |
| FDR | 0.41 | 0.47 | 1.37 | 9.34 | FDR | 0.40 | 1.38 | 1.32 | 12.86 |
| TPR | 1.37 | 7.69 | 1.53 | 1.23 | TPR | 0.47 | 0.95 | 1.42 | 0.70 |
| ACC | 0.20 | 1.04 | 0.36 | 3.08 | ACC | 0.10 | 0.26 | 0.28 | 5.10 |
| RMSE | 11.00 | 30.45 | 13.44 | 9.46 | RMSE | 5.51 | 7.12 | 13.00 | 6.56 |
| Test pMSE | 0.37 | 1.19 | 0.49 | 0.68 | Test pMSE | 0.10 | 0.05 | 0.24 | 0.29 |
| Train pMSE | 1.28 | 9.32 | 1.09 | 1.29 | Train pMSE | 0.20 | 0.33 | 0.56 | 0.45 |
| Case 3. $\boldsymbol{n=1000}, p=1600, \sigma_Y^2=1$ | | | | | Case 4. $n=600, p=1600, \boldsymbol{\sigma_Y^2=5}$ | | | | |
| | ST-CAR | ElasNet | STGP | T-LoHo | | ST-CAR | ElasNet | STGP | T-LoHo |
| FDR | 0.95 | 0.46 | 0.89 | 4.40 | FDR | 0.29 | 0.50 | 1.13 | 12.86 |
| TPR | 0.00 | 0.73 | 0.60 | 0.31 | TPR | 3.02 | 7.01 | 2.61 | 2.05 |
| ACC | 0.17 | 0.13 | 0.19 | 0.91 | ACC | 0.40 | 0.95 | 0.43 | 7.18 |
| RMSE | 4.27 | 6.53 | 12.04 | 5.83 | RMSE | 14.65 | 33.31 | 14.72 | 12.51 |
| Test pMSE | 0.07 | 0.06 | 0.32 | 0.23 | Test pMSE | 0.62 | 2.60 | 0.89 | 0.97 |
| Train pMSE | 0.17 | 0.50 | 0.47 | 0.36 | Train pMSE | 2.47 | 10.40 | 1.75 | 1.74 |
| Case 5. $\boldsymbol{n=6000}, \boldsymbol{p=10000}, \boldsymbol{\sigma_Y^2=5}$ | | | | | Case 6. $\boldsymbol{n=6000}, \boldsymbol{p=10000}, \boldsymbol{\sigma_Y^2=10}$ | | | | |
| | ST-CAR | ElasNet | STGP | T-LoHo | | ST-CAR | ElasNet | STGP | T-LoHo |
| FDR | 0.24 | 0.15 | 0.57 | 12.85 | FDR | 0.80 | 0.62 | 0.61 | 14.36 |
| TPR | 0.07 | 0.25 | 0.09 | 0.17 | TPR | 0.10 | 0.36 | 0.09 | 0.19 |
| ACC | 0.04 | 0.05 | 0.13 | 9.76 | ACC | 0.13 | 0.12 | 0.14 | 10.27 |
| RMSE | 2.26 | 3.00 | 6.43 | 2.30 | RMSE | 3.14 | 3.94 | 6.44 | 2.48 |
| Test pMSE | 0.18 | 0.23 | 1.40 | 1.54 | Test pMSE | 0.31 | 0.40 | 1.76 | 1.42 |
| Train pMSE | 0.32 | 1.33 | 1.58 | 1.88 | Train pMSE | 0.50 | 2.17 | 1.67 | 1.92 |

(b) IonS: standard deviations over 100 replicates. The evaluation criteria for estimation includes false discovery rate (FDR), true positive rate (TPR), overall accuracy (ACC), and root mean squared error (RMSE), all multiplied by 100. The evaluation criteria for predictive performance includes training and testing predictive MSE, denoted as Train and Test pMSE respectively

| Case 1. $n=600, p=1600, \sigma_M^2=5$ | | | | Case 2. $n=600, p=900, \sigma_M^2=5$ | | | |
|---|---|---|---|---|---|---|---|
| Criteria | ST-CAR | MUA | STGP | SBIOS | Criteria | ST-CAR | MUA | STGP | SBIOS |
| FDR | 1.49 | 1.55 | 1.42 | 1.49 | FDR | 2.16 | 2.44 | 2.35 | 2.12 |
| TPR | 1.31 | 1.45 | 0.88 | 1.25 | TPR | 3.26 | 3.11 | 2.01 | 3.04 |
| ACC | 0.36 | 0.44 | 0.42 | 0.35 | ACC | 0.66 | 0.68 | 0.63 | 0.57 |
| RMSE | 0.24 | 0.32 | 0.12 | 0.15 | RMSE | 0.28 | 0.35 | 0.25 | 0.22 |
| Case 3. $n=1000, p=1600, \sigma_M^2=5$ | | | | Case 4. $n=600, p=1600, \sigma_M^2=10$ | | | |
| Criteria | ST-CAR | MUA | STGP | SBIOS | Criteria | ST-CAR | MUA | STGP | SBIOS |
| FDR | 1.42 | 1.44 | 1.27 | 1.47 | FDR | 1.48 | 1.76 | 1.55 | 1.35 |
| TPR | 0.55 | 0.65 | 0.56 | 0.61 | TPR | 2.16 | 1.95 | 1.29 | 2.15 |
| ACC | 0.35 | 0.38 | 0.40 | 0.38 | ACC | 0.45 | 0.47 | 0.43 | 0.41 |
| RMSE | 0.16 | 0.20 | 0.07 | 0.08 | RMSE | 0.33 | 0.46 | 0.17 | 0.21 |
| Case 5. $n=600, p=6400, \sigma_M^2=5$ | | | | Case 6. $n=1000, p=6400, \sigma_M^2=20$ | | | |
| Criteria | ST-CAR | MUA | STGP | SBIOS | Criteria | ST-CAR | MUA | STGP | SBIOS |
| FDR | 0.73 | 0.74 | 0.79 | 0.77 | FDR | 0.88 | 0.68 | 0.90 | 0.65 |
| TPR | 0.80 | 0.89 | 0.28 | 0.75 | TPR | 1.22 | 1.23 | 0.41 | 1.19 |
| ACC | 0.21 | 0.25 | 0.26 | 0.21 | ACC | 0.29 | 0.26 | 0.28 | 0.22 |
| RMSE | 0.26 | 0.29 | 0.04 | 0.08 | RMSE | 0.28 | 0.32 | 0.06 | 0.12 |

probability(PIP)

$$PIP(\beta(s_j)) = 1 - P_j^0$$

where both `CAVI` and `SSVI` can trace the mixing probability $P_j^0$. We use the converged value at the last iteration of $P_j^0$ in `CAVI` to compute PIP. Since `SSVI` is a stochastic method, we use the averaged $P_j^0$ over the last 2000 iterations to compute its PIP. For the MCMC methods (`Gibbs`,`BIMA`), we

directly use the posterior sample of $T_\nu(\mu_j)$ (for `Gibbs`) or $\beta(s_j)$ (for `BIMA`) being nonzero over the last 20% of iterations as the posterior inclusion probability. For the final selection reported in Table A7, we use the true generating image $\beta$, and set a threshold $t$ on PIP: if $PIP(\beta(s_j)) < t$, $\beta(s_j) = 0$, otherwise $\beta(s_j)$ equals the posterior sample mean or the variational mean. By tuning $t$, we can control the FDR to be below 10%.

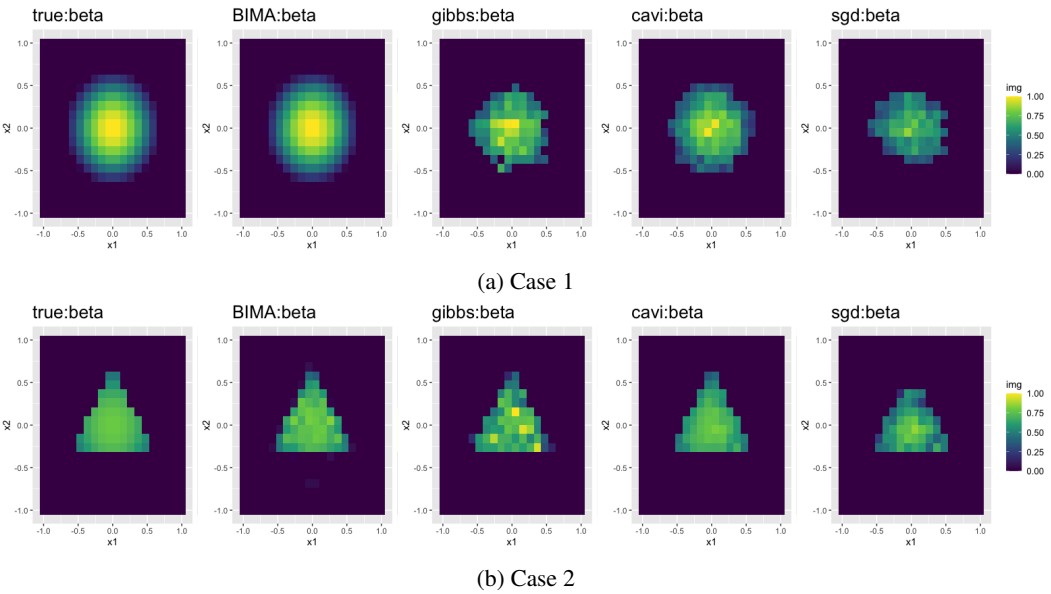

(a) Case 1

(b) Case 2

Figure A4: Illustration of estimated $\beta$ under 2 different cases.

Simulation I provides a relatively low-dimensional small-scale example, where $n = 200, p = 400$. We simulate two testing image cases for $\beta$ in equation 3, as shown in Fig A4a-A4b. In case 1, the true image intensity has a smooth transition from 1 to 0, and the voxels around the edge of the signal tend to have low signal-to-noise ratio, but the signal region is a smooth round shape that can be easily estimated by smooth Gaussian process. In case 2, the true image of $\beta$ is a sharp triangular shape, but the edge voxels of the signal has a sharp contrast to 0, with higher signal to noise ratio compared to case 1.

Table A7 provides the simulation results of the posterior mean estimates of $\beta$, with mean and standard deviation computed across 100 replications. SSVI has the best time efficiency across 4 Bayesian methods.

### A.10.2 SIMULATION A2: HIGH-DIMENSIONAL COMPARISON BETWEEN CAVI AND SSVI (IONS)

In the second simulation A2, we provide the additional to results to the simulation II IonS with a further comparison between CAVI and SSVI in high-dimensional settings. Table A8 provides the additional result on the performance of CAVI compared to SSVI.

Table A7: Simulation results based on 100 replications, with standard deviation in the bracket. All values are timed by 100 except for time (in seconds). FDR (false discovery rate) is the proportion of times that zero coefficients are identified as nonzero among all identified nonzero coefficients. Power is the proportion of times that nonzero coefficients are identified as nonzero among all nonzero coefficients. Accuracy is the proportion of times the prediction is correct. RMSE is the root mean square error over all voxels.

| Case1 | Gibbs | CAVI | SSVI | BIMA | glmnet |
|---|---|---|---|---|---|
| FDR | 5.4 (3) | 7.8 (2) | 7.8 (2) | 13.5 (1) | 3.0 (3) |
| Power | 80.0 (7) | 94.4 (3) | 84.9 (4) | 100.0 (0) | 24.1 (7) |
| Accuracy | 92.6 (2) | 95.9 (1) | 93.3 (1) | 95.3 (0) | 77.0 (2) |
| RMSE | 9.1 (2) | 5.4 (1) | 11.0 (1) | 0.5 (0) | 19.7 (3) |
| time | 97.7 (5) | 43.2 (10) | 12.7 (0) | 29.4 (1) | 1.2 (0) |

(a) Case 1

| Case2 | Gibbs | CAVI | SSVI | BIMA | glmnet |
|---|---|---|---|---|---|
| FDR | 8.0 (3) | 3.7 (0) | 2.0 (2) | 16.6 (3) | 0.0 (0) |
| Power | 100.0 (0) | 100.0 (0) | 97.0 (2) | 100.0 (0) | 94.7 (4) |
| Accuracy | 98.8 (0) | 99.5 (0) | 99.3 (0) | 97.4 (1) | 99.3 (1) |
| RMSE | 4.2 (1) | 1.9 (0) | 7.3 (1) | 2.2 (0) | 1.8 (1) |
| time | 101.4 (11) | 15.9 (5) | 12.9 (1) | 21.7 (1) | 1.2 (0) |

(b) Case 2

Table A8: Additional Simulation results to Simulation II. Comparison between CAVI and SSVI for ST-CAR prior, based on 100 replications.

| | FDR | | TPR | | ACC | |
|---|---|---|---|---|---|---|
| Case | SSVI | CAVI | SSVI | CAVI | SSVI | CAVI |
| Case 1. $n = 600, p = 1600, \sigma_M^2 = 5$ | 5.8 | 6.89 | 95.41 | 95.15 | 97.86 | 96.94 |
| Case 2. $n = 600, \boldsymbol{p = 900}, \sigma_M^2 = 5$ | 4.95 | 6.23 | 84.34 | 84.05 | 96.27 | 95.32 |
| Case 3. $\boldsymbol{n = 1000}, p = 1600, \sigma_M^2 = 5$ | 7.1 | 7.91 | 98.38 | 98.31 | 98.13 | 97.32 |
| Case 4. $n = 600, p = 1600, \boldsymbol{\sigma_M^2 = 10}$ | 5.07 | 5.64 | 85.31 | 84.79 | 96.06 | 95.85 |
| Case 5. $n = 600, p = 6400, \boldsymbol{\sigma_M^2 = 5}$ | 5.97 | 11.72 | 93.64 | 94.18 | 97.35 | 91.3 |
| Case 6. $n = 600, p = 6400, \boldsymbol{\sigma_M^2 = 20}$ | 5.93 | 8.57 | 81.83 | 83.58 | 95.06 | 91.73 |

| | RMSE | | Total time (seconds) | | Number of iteratios per second | |
|---|---|---|---|---|---|---|
| Case | SSVI | CAVI | SSVI | CAVI | SSVI | CAVI |
| Case 1. $n = 600, p = 1600, \sigma_M^2 = 5$ | 7.86 | 9.13 | 73.7 | 239.3 | 137.5 | 41.3 |
| Case 2. $n = 600, \boldsymbol{p = 900}, \sigma_M^2 = 5$ | 7.88 | 9.17 | 55 | 155.8 | 186 | 63.8 |
| Case 3. $\boldsymbol{n = 1000}, p = 1600, \sigma_M^2 = 5$ | 6.44 | 7.11 | 117.3 | 429.8 | 88.9 | 23.1 |
| Case 4. $n = 600, p = 1600, \boldsymbol{\sigma_M^2 = 10}$ | 10.32 | 12.7 | 82.9 | 255.4 | 122.7 | 39.2 |
| Case 5. $n = 600, p = 6400, \boldsymbol{\sigma_M^2 = 5}$ | 8.52 | 8.99 | 409.2 | 1282.8 | 24.7 | 6.7 |
| Case 6. $n = 600, p = 6400, \boldsymbol{\sigma_M^2 = 20}$ | 13.84 | 13.72 | 596.7 | 2641.5 | 17 | 3.4 |

