# OpenReview forum: "Bayesian Image Regression with Soft-thresholded Conditional Autoregressive Prior"
_ICLR.cc/2025/Conference — ICLR 2025 Poster_

### Official Review · Reviewer_NeTL · 2024-11-01

**Soundness:** 4
**Presentation:** 4
**Contribution:** 4
**Rating:** 8
**Confidence:** 4

**Summary:**

In this work, the authors present a novel prior for use in bayesian image regression from functional magnetic resonance imaging (fMRI). Motivated by limitations of previous priors which require the prior specification of a gaussian process kernel, which can lead to bias and other issues in the regression unless the true signal is known beforehand, the authors provide a novel prior which relaxes smoothness assumptions on functional parameters. The authors demonstrate that this novel prior autoperforms previous priors by a significant margin.

**Strengths:**

I find this work to be altogether very strong work with some omissions in the experimentation that should be easily addressed. At its core this work is well-motivated by intuition regarding the shortcomings of previous priors - the relaxation of the smoothness assumption is easy to grasp, and provides a firm foundation for the reader to follow the following, more dense theoretical work. I found no issues with the theoretical work, and appreciated the rigor taken by the authors.

The inclusion of substantial numerical and real-data experiments is a huge boon to this work. The authors do a substantial amount of experimental validation here, and demonstrate that this novel prior outperforms existing priors.

The authors also include a substantial appendix with additional work which I think further solidifies this as a strong paper.

Well done!

**Weaknesses:**

The only minor quibble I have with this work is the lack of standard deviations in most of the results. For example, in table 1 you average over 100 replications, but don't provide any standard deviation measures. These should be provided to fully solidify the performance of your approach over other methods.

**Questions:**

All of the questions I have regard the performance over multiple repetitions and would be addressed simply by providing those standard deviations. Every other question I had was substantially addressed in the appendix.

---

> ### Author Response · Authors · 2024-11-18
> **Response to Reviewer NeTL**
>
> We thank the reviewer for their encouragement and affirmation of our work. Your word has made my day!
> Due to the space limit, the presentation would look messy if we added the standard deviation with all the numbers in Table 1. In the revised appendix, we have added Table A5 of standard deviations in the revised manuscript corresponding to the means in Table 1(a) and Table 1A(a).

---

### Official Review · Reviewer_1CfH · 2024-11-02

**Soundness:** 3
**Presentation:** 3
**Contribution:** 3
**Rating:** 6
**Confidence:** 4

**Summary:**

This paper presents a Soft-Thresholded Conditional AutoRegressive (ST-CAR) prior for Bayesian regression in high-dimensional brain imaging data, demonstrating its advantages in capturing complex spatial dependencies and quantifying uncertainty. The authors validate the ST-CAR model's effectiveness through simulations and real data, while employing variational inference and stochastic subsampling to enhance computational efficiency.

**Strengths:**

1. The ST-CAR prior introduces a novel method to handle spatial dependencies in high-dimensional imaging data, where the coefficient is assumed to be smooth and sparse across their spatial domain.

2. The proposed framework for solving the problem achieves substantial improvements in computational speed, making it more feasible for high-dimensional applications.

3. The framework is validated on both simulated data and real fMRI data from the ABCD study, demonstrating its effectiveness in identifying significant brain regions associated with cognitive development.

**Weaknesses:**

1. The ST-CAR model involves several tuning parameters (e.g., the number of neighbors, thresholding parameter and decay rate), it seems that they require careful designed to achieve optimal performance.

2. The ST-CAR model effectively identifies significant regions of WM tasks in the ABCD dataset, but it seems that it cannot provide insights into broader brain circuits or networks, which are increasingly recognized as critical for understanding complex brain functions and disorders.

**Questions:**

1. Given the model’s focus on identifying significant brain regions, could it be extended to analyze inter-regional connections or brain circuits in cognitive functions?

﻿2. How sensitive is the ST-CAR model to different choices of the number of neighbors, the thresholding parameter and the decay rate? Do those hyperparameters stay robust in other datasets (e.g., HCP datasets)

3. Would it be possible to consider structural connectivity among regions of interest (extracted from DTI) as the exposures or confounders？

---

> ### Author Response · Authors · 2024-11-18
> **Response to Reviewer 1CfH**
>
> ## Response to Weaknesses:
> ### #1
> > The ST-CAR model involves several tuning parameters...
>
> This is similar to the weakness comment #1 from Reviewer RCFm.
>
> First, ST-CAR is robust to the hyperparameter settings as long as the thresholding parameter and the initial value of $\sigma_\beta$ are in a reasonable range to reflect the size of the actual signals. This is evidenced by our extensive simulation studies shown in both the SonI and IonS cases, where the true signal patterns vary quite differently, but the hyperparameters are all fixed.
>
> Second, the sensitive and noisy result of SonI in the ABCD data analysis is not the fault of ST-CAR, but because the SonI itself is very high-dimensional with very small signals, as shown in the Elastic Net result in the revised Section A.7. This sensitivity happens to any methods on such a challenging problem. In fact, even in such a case, ST-CAR is still robust to the choice of number of neighbors and decay rate, as shown in Tables A2 and A3. The only sensitive parameters are $\nu$ and the initial value of $\sigma_\beta^2$, which is due to the small effect size. Basically, if the true effect size is on the scale of 0.0001, setting the thresholding parameter to be 0.1 would obviously be wrong. Both of these two hyperparameters are related to the signal size, not the smoothness structure. Hence our claim that ST-CAR is robust to varying smoothness of signal patterns without tuning the prespecified correlation structure still holds. The IonS sensitivity analysis is very stable across a range of hyperparameters.
>
> ### #2
> >The ST-CAR model effectively identifies significant regions of WM tasks in the ABCD dataset...
>
> The generalizability of ST-CAR to other brain data modality is not a weakness of ST-CAR, but a potential future direction. In fact, the only assumption ST-CAR makes is that the functional parameter needs to be sparse and smooth functions. If the brain circuits or network data have such features, then ST-CAR can no doubt be applied to other data types as well. In fact, due to the CAR structure which intrinsically considers each location as a grid points, we feel that generalizing ST-CAR to network data should be an easy task, especially given that ROI-based brain network analysis usually contains much less locations than fMRI data, which makes it a high SNR problem that is easier to make inference than the high-dimensional low SNR fMRI data.
>
> ## Response to Questions:
> ### #1
> See our response to Weakness #2.
>
> ### #2
> We have conducted thorough sensitivity analyses on all the aforementioned hyperparameters in Appendix A.6, and provided validation of the real data results using baseline methods Elastic Net and MUA in the revised Appendix A.7. In short, ST-CAR is not sensitive to these hyperparameters, but in the extreme case with very low SNR such as the SonI in ABCD data, the thresholding parameter and initial value of $\sigma_\beta^2$ needs to be carefully chosen to reflect the range of the signal size.
> In terms of SNR, SonI should have the lowest SNR due to the high-dimensionality and small signal sizes. The HCP data and other brain network data usually contain much less (a few hundreds ROIs compared to 40k+ voxels), hence ST-CAR should be much more robust to network data regression than imaging data regression.
>
> ### #3
> Yes. For exposure variables, the current implementation is sufficient. To add additional functional confounders, one only needs to expand ST-CAR to the corresponding coefficients of confounders.

---

> > ### Comment · Reviewer_1CfH · 2024-11-20
> >
> > Thanks for your reply. I have decided to change my rating to 6. It is a well-designed study. I hope the complementary about other exposure variables will add to your final submissions.

---

### Official Review · Reviewer_85HH · 2024-11-03

**Soundness:** 3
**Presentation:** 3
**Contribution:** 3
**Rating:** 6
**Confidence:** 4

**Summary:**

This paper introduces a novel prior, the Soft-Thresholded Conditional AutoRegressive (ST-CAR) prior, for handling high-dimensional imaging data in regression models, specifically for brain imaging studies. The ST-CAR prior adapts to data-driven spatial smoothness, controlling sparsity, and incorporating inclusion probabilities for selecting active voxels.

The ST-CAR prior is applied to two common regression models in neuroimaging: scalar-on-image (SonI) and image-on-scalar (IonS). In SonI, brain imaging data are used to predict scalar outcomes, such as cognitive scores, while in IonS, scalar predictors (e.g., parental education level) are used to predict imaging outcomes. The paper introduces variational inference (VI) algorithms, including Coordinate Ascent Variational Inference (CAVI) and Stochastic Subsampling VI (SSVI), to make the approach computationally feasible for large-scale datasets. Results from simulation studies and a real-data analysis using the Adolescent Brain Cognitive Development (ABCD) study demonstrate that the ST-CAR prior outperforms traditional methods in identifying active brain regions and offers improved computational efficiency.

**Strengths:**

The authors have made a comprehensive presentation of their method and the presented results seem convincing. They have implemented both synthetic experiments and also a real ones, which strengthens their method. Overall, I believe that this paper can have an impact on the neuroimaging domain and making inferences on high-dimensional imaging data in curated clinical studies.

**Weaknesses:**

### Weaknesses
1. Dependence on Thresholding Parameter v.
- While the paper claims that the ST-CAR prior is relatively insensitive to the choice of the thresholding parameter v, the results could still be impacted by this setting. A suboptimal threshold could lead to either excessive sparsity (losing important signal) or insufficient sparsity (resulting in noise). Without clear guidelines for tuning v, users may struggle with finding the right balance, especially across different datasets. I would suggest to implement a sensitivity analysis showing how different values of v affect the results, or propose a data-driven method for choosing v.

2. Limited Applicability for High Signal-to-Noise Ratios (SNR)
 - The paper notes that the ST-CAR prior performs well in low-SNR settings, particularly with SonI models. However, in settings where SNR is high, the method might not be optimal compared to other methods that handle high-SNR data more effectively. This could limit the generalizability of the model to other types of imaging data or modalities with higher SNR.  Can you discuss potential challenges or modifications needed to apply ST-CAR to noisier, lower quality data. Have you tested or plan to test the method on less processed datasets? Can you propose specific preprocessing steps or model adjustments that might be necessary for such data?

**Questions:**

### Questions

- Can you discuss how this method could be applied to other imaging modalities, such as structural MRI, DTI, and PET imaging? Would any assumptions need to be adjusted when handling these modalities instead of fMRI data?

- Is it necessary to apply IonS and SonI regressions in such high-dimensional spaces? Are there atlas-based methods that could produce ROI-level features that are more interpretable? By using such features, one might make inferences at a more aggregated level. So, my question is: is it essential to perform voxel-level inferences to determine the relationships between brain ROIs and clinical outcomes in such high-dimensional data? I would like the authors to elaborate on the need to work in this high-dimensional space.

- Clinical study data are often curated and carefully processed. Do you believe that ST-CAR could handle real imaging and clinical data that are frequently noisy and of lower quality?

---

> ### Author Response · Authors · 2024-11-18
> **Response to Reviewer 85HH**
>
> ## Response to Weaknesses
>
> ### #1
> >Dependence on Thresholding Parameter v.
>
> We thank the reviewer for this suggestion, and note this suggestion has in fact already been included in the manuscript and the appendix. We used a data driven way (2-fold cross-validation, appendix Section A.6 Table A2) to select $\nu$, and presented the sensitive analysis in the appendix Section A.6 Table A3, A4. ST-CAR is insensitive to the different smoothness of the true signals. But the sensitivity on the thresholding parameter largely depends on the scale of the effect size, and is a common problem to all other baseline methods. In Table A2, we use cross-validation to provide the reasonable range of the effect size and determine $\nu$ based on best testing pMSE. The revised Section A.7 also shows that baseline methods such as Elastic Net would have similarly small effect sizes for SonI, and the MUA result for IonS provides a sanity check on the regions identified by ST-CAR.
>
> ### #2
> >Limited Applicability for High Signal-to-Noise Ratios (SNR)
>
> Based on our simulation results in Table A1, ST-CAR outperforms baseline methods in all but cases 2 and 6 in terms of RMSE of $\alpha$ estimation, and in cases 2 and 6 where STGP has a lower RMSE, the FDR is above 10%, which is a sign of poor selection accuracy. Based on the empirical performance, we are confident that ST-CAR performs well in high SNR settings.
> Nonetheless, in the extremely low SNR setting such as the ABCD data SonI analysis, where p is much larger than n, and the signal sizes are below 0.01, it can be very challenging for ST-CAR, and all other baseline methods. The only other method that performs well in the low SNR setting is T-LoHo, but T-LoHo cannot generalize to very high-dimensional problems such as SonI with imaging data. To stabilize the performance of ST-CAR in the extremely low SNR case of the SonI data analysis, as discussed in our previous answers, we use a random training and testing split to select thresholding parameters that can reflect a reasonable range of the true signal sizes.
>
> ## Response to Questions
> ### #1
> >Can you discuss how this method could be applied to other imaging modalities...
>
> The only assumption needed is that the true signal is a sparse and smooth function. Any other data modality that satisfies this constraint can be applied using ST-CAR.
>
> ### #2
> >Is it necessary to apply IonS and SonI regressions in such high-dimensional spaces? ...
>
> The atlas-based method still relies on prespecified region parcellation. These region parcellations may contain prior knowledge about the brain anatomical structure, but oftentimes the signal pattern may not align with these prespecified region parcellations. Being able to perform voxel-level inference is definitely a plus, because ROI-based analyses contain much less locations (much smaller p), higher SNR, and are generally much easier to make inferences about than voxel-level analyses. ST-CAR can in fact be applied to ROI-based analysis easily. However, ROI-based inference can only reveal much less information than voxel-level analysis.
>
> ### #3
> >Clinical study data are often curated and carefully processed...
>
> The ABCD data analysis provides a good example on how ST-CAR can be applied to such noisy data. Based on the results for the real data analysis, ST-CAR provides stable interference results in IonS which can be validated by the MUA-discovered regions. In the extremely low SNR case such as the SonI data, ST-CAR may have produced a noisy estimation, but this is a common challenge for all baseline methods, as validated by the Elastic Net result in our revised Appendix Section A.7. If the data quality is too low and the signal sizes are also very small, there is very little  hope in meaningful analysis. But ST-CAR still provides a quick and scalable inference approach to do some basic explorative inference in this case.

---

### Official Review · Reviewer_RCFm · 2024-11-04

**Soundness:** 2
**Presentation:** 3
**Contribution:** 2
**Rating:** 3
**Confidence:** 4

**Summary:**

The authors propose a new Bayesian method for sparse regression. To solve the sparse regression settings, they introduce a method based on stochastic subsampling and variational inference. They verify their model using simulation data, and finally apply their model on fMRI task activation maps to predict the effect of brain regions on IQ development and how the task activation map is affected by parental education level. Throughout the paper the authors consider two regression settings; Image on Scalar (IonS) regression and Scalar on Image regression (SonI).

**Strengths:**

I think the authors set up an important and interesting problem both in their simulated setting and for fMRI data. Specifically, I believe that encoding spatial relationships in both regression settings is important, and I think the significance of the authors’ method can therefore be high. The authors compare their method to good baselines, and I think the comparisons are relatively fair. The idea is original and builds on the soft-thresholded Gaussian process, which the authors cite and compare against. I think the authors also do a relatively good job of explaining their method, and their results on simulated data are good. However, the clarity of the method can be improved.

**Weaknesses:**

Major weakness(es): \
I think the two main major weaknesses of this work are 1) that the paper lacks a thorough hyperparameter search and the method is very sensitive to hyperparameters and 2) that the fMRI analysis lacks depth.
1. First, authors describe a lot of hyperparameters on Pages 6 and 9, but none of these hyperparameters are explained or searched over. Specifically on Page 6 for the STGP model: a=0.01, b=10, 10 degrees of Hermite polynomials per sub-region, 10^4 iterations, thresholding parameter is 0.2. For ST-CAR: thresholding parameter, decay rate, number of neighbors, correlation parameter, and fixed variance parameter. The same thing applies for Page 9. It is of course not necessary to search over all of these hyperparameters, but it’s good to explain why you set them and search over the most important hyperparameters for competing methods. Especially since hyperparameters may make a big difference. Second, the method is very sensitive to its hyperparameters, as can be seen Table A2 &A3, and the training and test R^2 values do not correlate with each other. Specifically, the best training R^2 in Table A.2 (0.84 R^2) achieves the worst test R^2 (0.12). If the authors had thus used the best training model, they would have performed worse than ElasticNet and STGP on the held-out test set (0.23 and 0.28 R^2 on the test set respectively). The same behavior is found in Table A3, where the best test R^2 (0.64) is achieved at the worst training R^2 (0.22) and vice-versa (0.51 vs 0.41). In fact this to me puts into question the simulated results, and I would strongly encourage the authors to perform a hyperparameter search, pick the best hyperparameter based on validation set results (i.e. separate the data into a training, validation, and test set), and then evaluate each model (including their own) with those hyperparameters on the test set. I specifically mention this point as well because the authors select the best v and $\sigma_\beta^2$ based on the best testing performance (L443), which is bad practice, since it can lead to overfitting. Especially since there is almost an inverse relationship between the training and test performance (as described previously).
2. The authors mention throughout the paper that many of the assumptions they make in their model hold for fMRI data (L162 for example). Hence, I would assume this paper to have more of a focus on actually showing why this model works well for fMRI data. However, I think the fMRI analysis lacks depth, and the results in Figure 3a do not look convincing because of how noisy the significant regions look. The noisiness is underlined by the results in Table A2: the IQ predictions do not seem to generalize well from the training to the test set. Additionally, the authors do not show whether their method actually identifies different areas than any of the baseline methods, which would help verify that their model is a helpful tool for the neuroimaging field compared to the baseline methods. Although I understand that the authors choose not to go too in-depth about the results since this is a machine learning conference, I think given the focus on fMRI in this paper the authors should expand a lot on their current discussion of what these results indicate (i.e. why does it make sense that these regions are related to IQ and parental education level?), atleast in the Appendix. Lastly, and this is not as important to address, but I think there are additional potentially more interesting questions one can ask using the ABCD dataset, i.e. regressions to mental health assessments [1].

Minor weaknesses:
1. The authors mention that there is a sparse signal in fMRI data, the authors should expand on this claim. Similarly, on Line 162 the authors claim that the signal is assumed to be sparse and piecewise smooth, but the authors should expand on this claim and potentially provide references. I agree generally with the statement, but I do think it’s important for the authors to explain why.
2. The notation the authors use can be confusing, the authors use $I_d$ for an identity matrix of dxd dimensions, but also use $I(\cdot)$ as the indicator function. I would encourage the authors to use $\mathbb{1}_{\\{\cdot\\}}$ as notation for the indicator function instead. Similarly, instead of using $\mathcal{N(j)}$ for a neighborhood and $\mathbf{N}(\cdot)}$ for a Normal distribution, I would use $\mathcal{N}$ for a Normal distribution and other notation for the neighborhood. Lastly, $p_j$ is not defined in the text on Line 131.
3. The authors do not compare their model with T-LoHo on the data that is used in the T-LoHo paper. I assume because ST-CAR cannot handle non-smooth spatial flips from positive to negative values well (i.e. Line 161), but the authors do not mention this in the limitations section or show any results on these types of edge cases.
4. The authors should also and more importantly report FLOPs instead of computation time because computation time is machine-dependent.
5. The authors mention fMRI pre-processing  on Line 422, but do not explain in the paper what types of pre-processing they perform. Moreover, they do not explain how they select the subjects they use and how they split the data into a training and test set.
6. The authors only show the positive effects in their fMRI results (Figure 3) and Table 2, which they should do, since it isn’t clear why only positive effects are expected for the hypotheses they test.
7. The quality of Figure 3 is not up to par, it is hard to read the color bar numbers.

Spelling/grammar:
- L36-37: “…, and it can be time-series …, or in our particular interest, the functional Magnetic Resonance Imaging data … of human brain signal.” -> .This high-dimensional component can be a time-series, …, or in our case, functional magnetic resonance imaging (fMRI) data of the human brain.
- L39/40: “…, formulated as a scalar-on-image (SonI) regression…” -> formulated as scalar-on-image (SonI) regression
- L54/55: “… is still computationally expansive...” -> is still computationally expensive
- L109/109: “…, referred to as the…” -> called the
- L247: “… algorithm, but cannot directly give inference results …” -> algorithm, but cannot directly infer
- L271/272: “… selection accuracy in SonI setting.” -> selection accuracy in the SonI setting.
- L305: “…, we use ridge regression result …” -> we use the ridge regression result
- L371/372: “… blow 10% …” -> below 10%

[1] Chen, J., Tam, A., Kebets, V., Orban, C., Ooi, L. Q. R., Asplund, C. L., ... & Yeo, B. T. (2022). Shared and unique brain network features predict cognitive, personality, and mental health scores in the ABCD study. Nature communications, 13(1), 1-17.

**Questions:**

1. L102: I don’t quite understand why the authors say that the function on L101 shrinks values below $v$ to zero, since $I(\cdot)$ is the indicator function isn’t any value below $v$ zero?
2. Based on Figure 1 I was wondering whether it’s not just possible to use a different kernel to get better results both on the simulated data, i.e. reducing the variance in the exponential kernel or using a kernel like the Jump Gaussian process [1]? The authors select hyperparameters for the STGP model on L288, but do not provide any reason for these hyperparameters. I would strongly encourage the authors to do a thorough hyperparameter search on a validation set for all of the methods they compare against.
3. On Line 213/214 the authors mention the design matrix M, but isn’t M the task activation map? The design matrix normally contains regressors of interest and nuisance regressors.
4. Have the authors looked at how initialization affects each model? I understand there are differences that may make this hard for some methods, but ST-CAR is initialized differently than the other methods.
5. Why did the authors choose to look at IQ and parental education level effects in the brain? And why did the authors use the 2-back task activation map?

[1] Park, C. (2022). Jump Gaussian process model for estimating piecewise continuous regression functions. Journal of Machine Learning Research, 23(278), 1-37.

---

> ### Author Response · Authors · 2024-11-18
> **Response to Reviewer RCFm**
>
> Thank you for your detailed comments and suggestions. We would like to clarify that the main goal of this paper is not to predict new outcomes, but rather to make inferences and identify important brain regions that contribute to the association between exposure and outcome. We acknowledge that many models may be more accurate for prediction than linear regression. Instead,  our focus is on the linear model due to its straightforward interpretation and robust performance. We propose a new prior model that can be flexibly adaptive  to varying spatial patterns, and it leads to efficient posterior computational algorithms such as CAVI and SSVI. The SonI and IonS regression models serve as examples to illustrate the strength of this prior.
>
> Regarding the sensitivity of the SonI results, we believe this is not caused by the proposed ST-CAR method. Instead, it is due to the low SNR of the SonI problem itself, as evidenced by the Elastic Net baseline result in Section A.7 of the revised manuscript. In contrast, for high SNR IonS problem, ST-CAR IonS shows stable performance across hyperparameters.
>
> # Response to Weaknesses:
> ## Major weakness #1 (part 1)
> > First, authors describe a lot of hyperparameters on Pages 6 and 9, but none of these hyperparameters are explained or searched over...
>
> First, we acknowledge that the hyperparameters play an important role in all the baseline methods and ST-CAR.  The hyperparameters for ST-CAR are thoroughly explained in Section 2.2, and our choices are designed to ensure fair and meaningful comparisons in the simulation studies. Specifically, we selected hyperparameters to provide reasonable and consistent estimation of the true patterns across all methods, without conducting an exhaustive search for optimal values. The goal is to show that ST-CAR performs better than the existing other methods under varying shapers and smoothness of the true signals for one set of hyperparameters.
>
> For instance, the thresholding parameter for STGP and ST-CAR was chosen based on  the true signal sizes, which range between -1 and 1. Setting a threshold of one would be inappropriate. The choice of $\nu$ in ST-CAR is explained on Line 306-308. Similarly, other hyperparameters were selected to ensure reasonable performance on simpler ball-shaped signals. For the other three more complex shapes, we maintained the same hyperparameters to evaluate the robustness of  each method to the different levels of signal smoothness .
>
> The exponential square kernel for STGP is a reasonable choice given the scale of the support ([-1,1]^2) and the true signal pattern (shown in Figure 2 and A1). Figures 2 and A1 indicate that the chosen GP kernel can at least identify the active area to a certain extent, for example, in Figure 2, STGP obviously estimates the ball-shaped pattern well, whereas its estimation on the M-shape can be improved. While fine-tuning the GP kernel hyperparameters could potentially enhance STGP's performance, the computational cost of such tuning for each new signal pattern would be prohibitive.
>
> We acknowledge that hyperparameter tuning can improve the performance of both ST-CAR and other baseline methods, but our primary focus is on demonstrating robustness without exhaustive tuning. The diverse  shapes in Figure 2 and A1 highlights that even without  hyperparameters tuning, ST-CAR is capable of aptively identifying different patterns effectively. Our goal is to present a novel method that remains robust across different signal patterns without requiring extensive hyperparameter turning, and we believe we have demonstrated that ST-CAR can achieve this goal with a fixed set of hyperparameters. This careful selection of hyperparameters ensures that our comparisons in the simulations are fair and that the reported performance differences are meaningful.
>
> >Second, the method is very sensitive to its hyperparameters, as can be seen Table A2 &A3...
>
> Second, regarding the real data analysis, we acknowledge that the SonI result can be sensitive to the threshold parameter $\nu$ and the initial value of $\sigma_\beta^2$. However, this sensitivity is not  an inherent problem with ST-CAR, but is rather a challenge  posed by  the small effect size of the fMRI data in the SonI problem. In contrast, the IonS problem exhibits  more stable performance in terms of training and testing pMSE (Tables A2(b) and A4)  across varying hyperparameters and has more contiguous active regions (Figure 3(b)). This stability is because the data for the IonS analysis has a much higher SNR with a larger scale compared to SonI. Specifically, the visible range of positive active signals for SonI are between [0.0005, 0.001], whereas the range of IonS signals are in [0.05, 0.1]. The smaller scale of the SonIsignals require a careful selection of the threshold parameter $\nu$ and the initial value of $\sigma_\beta^2$. (to be continued)

---

> ### Author Response · Authors · 2024-11-18
> **Response to Reviewer RCFm**
>
> (... continued from the previous comment)
>
> This sensitivity issue is not related to the smoothness pattern but instead reflects the need for a reasonable understanding of the true signal effect size to determine an appropriate threshold. For example, if the true signals have effect sizes in [0.001, 0.01], setting a threshold at 1 would be inappropriate. Since $\sigma_\beta^2$ controls the difference between $\beta$ and the soft-thresholded $\mu$; and thus its  initial value also influences the estimations on the effect size of $\beta$.  In fact, the analysis in Table A2 was conducted to estimate the potential range of the signal sizes.  Further sensitivity analysis in Table A3 shows that ST-CAR is not sensitive to the bandwidth and the decay rate.
>
> We respectfully disagree with the statement that “the training and test R^2 values do not correlate with each other”. As indicated in Table A2(a),  there is indeed a negative correlation between training and test R^2 values, as the reviewer mentioned that “the best training R^2 in Table A.2 achieves the worst test R^2”. This is exactly the evidence for overfitting. Overfitting occurs when   the threshold is too small. For example, with $\nu$=0.003, ST-CAR can achieve the best train pMSE, which results in noise from the training set picked up as signals as the soft-thresholding fails to  eliminate these noise elements. This underscores the importance of identifying  a reasonable range for the signal size  especially for challenging problems such as SonI, to avoid overfitting. This is a problem faced by all baseline methods. This is why methods like Elastic Net need cross-validation to determine the threshold parameters.
>
> Within our computational power, we have conducted a sanity check using Elastic Net and MUA separately for SonI and IonS to validate that the signals identified by ST-CAR are reasonable. The results are shown in the revised version Appendix Section A.7. Based on the new Figures A2 and A3 in comparison with Figure 3, we observe that IonS has  stable results, and the regions identified by MUA are  more noisy,  lacking a reasonable BH-adjusted p-value for uncertainty quantification. The result for SonI obtained by Elastic Net further verifies the extremely small effect size. This further supports our claims that the relatively sensitive result of ST-CAR in the real data analysis is not a method limitation, but rather a common problem faced by all methods when SNR is very low. From our experiments in simulations, ST-CAR achieves better performance compared to other methods in identifying signals under such low SNR settings. In addition, we believe there may be a misunderstanding regarding our approach. The reviewer mentioned that “selecting hyperparameters based on testing MSE leads to overfitting”. But  it is well known in the machine learning community that overfitting is indicated by a low training MSE  and a high test MSE. Using the test MSE to choose the hyperparameters is to avoid overfitting. We would like to clarify our primary objective here, which is not focused on evaluating prediction performance but rather on estimating model fitting R-squared. The test R-squared reported in our current work serves as a correction to the training R-squared, intended to provide a more realistic assessment of model fit. As such, our interest lies in evaluating the fit of the model to the underlying data rather than estimating the prediction accuracy.
>
> We appreciate all your insights, and we hope our clarifications address your concerns regarding the approach we have adopted in our analysis.
>
>
> ## Major weakness #2
>
> >The authors mention throughout the paper that many of the assumptions they make in their model hold for fMRI data...
>
> We would like to clarify that  our model relies on  one key assumption:  the brain imaging signals from fMRI data are sparse and smooth functions. The statements on Line 162 is a  direct result  of this smoothness assumption, i.e. a smooth function by definition does not have discontinuous jumps from positive to negative values. The sparsity and smoothness assumption is well recognized in the neuroimaging community (see [2,3,4]). To make this clear we have added a sentence with references on this assumption in the revised manuscript Line 46-47.
>
> Regarding the noisiness of   for SonI in Figure 3(a), as we previously mentioned in response to comment #1, SonI is a challenging problem when  SNR is low. We have included Section A.7 in the revised appendix to present  a separate analysis  of SonI and IonS using Elastic Net and MUA. Table A2  presents a reasonable range of the signal size, which is also validated by the Elastic Net result. We have also added the thresholded $\beta$ value in the revised Figure A3, which shows ST-CAR still tends to identify more activation voxels than ElasNet, and some activation regions still overlap on the sagittal plane (first plot from the right) with ElasNet result. (TBC....)

---

> ### Author Response · Authors · 2024-11-18
> **Response to Reviewer RCFm**
>
> (... continued from the previous comment)
> For the interpretation of the identified brain regions, we discussed the cognitive function of a few discovered regions on Line 485, Line 505-510. These discussions provide insights into why the identified regions are relevant to IQ and parental education level, offering a more in-depth understanding of the results.
>
> We acknowledge that other, potentially more interesting, questions could be explored using the ABCD dataset, such as regressions involving mental health assessments. We agree that these are promising directions, and exploring such applications would be an interesting avenue for future work to extend the applicability of our proposed ST-CAR model.
>
> ## Minor weaknesses:
> ### #1
> We thank the reviewer for this comment, and have added a sentence to provide references on this assumption in the revised manuscript Line 46-47 in the introduction section. As mentioned above, Line 162 is a natural result due to the smoothness.
>
> ### #2
> We thank the reviewer for this helpful suggestion. Although we use the same letter I for Indicator function and Identity matrix, but the fonts are different, it should not impair the readability based on the context. Similarly for Normal distribution and Neighborhood using the letter N but with different math fonts. In addition, the notations are all introduced in section 2.1 for the readers to check. We may consider changing the notation as the reviewer suggested if our paper is accepted.
>
> ### #3
> We appreciate the reviewer’s comment. We explicitly mention the limitation of ST-CAR on Line 161, noting its inability to effectively handle non-smooth spatial transitions from positive to negative values.  Given the page limits, we chose not to include a separate section for limitations. Furthermore,  T-LoHo is designed for network data with clustering structure, which is not suitable for the fMRI data with spatially varying smooth functional patterns, as shown in Figure 2 where the ball-shape and M-shape present clear clustering structure with discrete functional values. “All models are wrong, but some of them are useful.” Expecting one method to do the best in all scenarios is unrealistic. Hence we only advocate ST-CAR as a suitable approach for imaging regressions involving spatially smooth patterns.
>
> ### #4
> We thank the reviewer for this suggestion. FLOPs might be a more accurate way of presenting the computational time. However, the current computational time results are performed on the same HPC cluster with random core assignment. We think this result is sufficient to demonstrate the superiority of ST-CAR’s VI algorithms over competing methods. Using FLOPs may have marginal differences, but it will not largely impact the comparison results.
>
> ### #5
> The preprocessing method is cited on Line 419 (Sripada et al., 2020). The subjects are samples from ABCD Release 1.0 data (Line 418), and the training and testing is based on a random split of 70% to 30% (Line 444). We have added more details and references in the revised Section 5.
>
> ### #6
> We use the Posterior Inclusion Probability (PIP) as the criterion for uncertainty quantification. Based on Figure 3 and Table 2, after applying thresholds of PIP>0.25 for SonI and PIP>0.95 for IonS, no significant negative voxels were identified (Line 484-485).
>
> ### #7
> We apologize for the visibility of the color bar in Figure 3. However, this is a problem with the software MRIcron for visualizing fMRI data. Instead, we have made an effort to explain the color bar with the captions in Figure 3, and in the main text Line 477-479.
>
> ### Spelling/grammar
> We thank the reviewer for the careful reading. All have been corrected.
>
> ## Response to Questions
> ### #1
> This sentence simply means that the soft-thresholding function $T_{\nu}(x)$ shrinks $x$ values where $\text{abs}(x)<\nu$ to be 0.

---

> > ### Author Response · Authors · 2024-11-18
> > **Response to Reviewer RCFm**
> >
> > (... continued from the previous comment)
> >
> > ## Response to Questions
> > ### #2
> > As explained in the introduction section for the motivation of this paper, the performance of many GP kernels rely on prespecified kernel structure. Similarly, PCA based methods rely on the specific basis, MRF-based methods rely on the neighborhood structure, etc. This dependence on prespecified structure is not unique to GP, but a general phenomenon. Of course, optimizing over the kernel parameters can get better performance, but this is inconvenient and computationally infeasible for large-scale data such as fMRI data. This inconvenience motivates us to develop a method that is robust to such prespecified smoothness structure. As mentioned previously, the focus is not on the hyperparameter search for STGP to gain better performance, but to design a method that is insensitive to different signal patterns. The kernel for STGP is chosen so that it can reasonably estimate the signal pattern as shown in the simulation studies. The estimations of STGP are indeed reasonable, they are just not sharp enough to adapt to different patterns. This is exactly what ST-CAR can do, and STGP and other baseline methods cannot. The point is to relax the dependence on the prior smoothness assumption.
> >
> > ### #3
> > In the SonI linear regression, M is the design matrix for beta. Of course, the combination (M,X) of dimension n by (p+q) is the design matrix for (beta, gamma) jointly. In the neuroimaging application, M is the task fMRI data.
> >
> > ### #4
> > The initialization needs to be reasonably dense for ST-CAR in a low SNR case such as SonI regression. For example, if we had initialized $\beta$ to be all 0, it would be challenging for CAVI to escape local optimum and achieve global optimum. Again, this is due to the optimization nature of CAVI, and when we experimented with ST-CAR using Gibbs sampler, the initial values would not make a difference. Hence in the simulation and real data analysis, we used the ridge regression result as the initial value, which by itself is a data-driven way and is easy to implement. For other computational methods such as MALA used in STGP, we used Elastic Net as the initial value to avoid long exploration periods due the slow convergence of MALA. Hence ST-CAR is not the only method that is initialized differently, we have made initial values suitable for each method. The reasoning for choosing such initial values is explained in Line 305-309.
> >
> > ### #5
> > Thank you for this helpful suggestion.  We have added a section A.8 in the revised appendix to provide scientific evidence for the ABCD analysis motivation. “Recent studies (Cermakova et al., 2023; Halabicky et al., 2023) have demonstrated that parental education levels are significantly associated with children’s cognitive abilities, including specific cognitive functions such as working memory. In particular, children’s cognitive functions are often associated with their brain development, which fMRI captures during tasks like the working memory task in the ABCD study.“ The 2-back task fMRI data can better reflect the working memory, as explained in [5].
> >
> > ## References
> > [1] Park, C. (2022). Jump Gaussian process model for estimating piecewise continuous regression functions. Journal of Machine Learning Research, 23(278), 1-37.
> >
> > [2] Kang, Jian, Brian J. Reich, and Ana-Maria Staicu. "Scalar-on-image regression via the soft-thresholded Gaussian process." Biometrika 105.1 (2018): 165-184.
> >
> > [3] Taylor, Jonathan E., and Keith J. Worsley. "Detecting sparse signals in random fields, with an application to brain mapping." Journal of the American Statistical Association 102.479 (2007): 913-928.
> >
> > [4] Smith, Michael, and Ludwig Fahrmeir. "Spatial Bayesian variable selection with application to functional magnetic resonance imaging." Journal of the American Statistical Association 102.478 (2007): 417-431.
> >
> > [5] Chandra Sripada, Mike Angstadt, Saige Rutherford, Aman Taxali, and Kerby Shedden. Toward a “treadmill test” for cognition: Improved prediction of general cognitive ability from the task activated brain. Human brain mapping, 41(12):3186–3197, 2020.

---

> ### Author Response · Authors · 2024-11-26
> **Response to Reviewer RCFm**
>
> We appreciate the reviewer’s feedback and understand the concerns about the noisy pattern in the SonI results. While it is true that the ABCD dataset might not showcase an optimal signal-to-noise ratio (SNR) for this specific task, the ABCD fMRI data forms the core motivation for our work. This dataset is uniquely suited to addressing critical scientific questions in neurodevelopmental research due to its large scale, rich demographic diversity, and comprehensive multimodal measurements. Our aim is to develop methods that can handle the real-world challenges posed by such complex, large-scale data, including noise and the absence of ground truth, as these are representative of the data scenarios researchers are most likely to encounter. By focusing on the ABCD dataset, we prioritize methodological relevance and practical applicability over purely performance-driven comparisons on artificially clean datasets. This approach aligns with our broader goal of advancing methods that are robust and effective for meaningful scientific inquiry.
>
>
> For the purpose of comparing ST-CAR with other methods on a real data experiment, the IonS result with ABCD fMRI data has served this purpose and shows a promising and stable estimation result. Comparing Figure 3(b) and Figure A2 (b), we can clearly see how the ST-CAR prior is able to borrow information across the space to reduce false discoveries, especially compared with the MUA result which ignores spatial structures. Hence the IonS experiment can show that we have “evaluated our method on real data with some reasonable ground-truth and show how hyperparameters affect the performance of our method”, as requested by the reviewer.
>
>
> For the SonI experiment with ABCD fMRI data, none of the existing methods other than Elastic Net can be applied to such a scale of problem within a reasonable time, and STCAR still achieves a better test MSE and R^2 compared to Elastic Net with cross-validation (Table A2(a)). It is important to highlightthat SonI is a particularly challenging problem due to the very small effect size. This necessitates borrowing strength across space and creating a novel method to tackle it effectively.
>
>
> To address the reviewer’s concern that the identified voxels in SonI might be due to random noises, we have conducted a stability check: we rerun the SonI on a slightly reduced subset of the ABCD data, by randomly removing 10% (or 30%) of the data to re-train the model, and check if the identified activation voxels in the full sample can still be recovered in each random subsample. We have conducted a total of 50 splits, and report the results in the following tables.
> In the tables below, we combine the identified voxels by ST-CAR and ElasticNet in the full sample analysis to serve as the “ground truth”. The numbers in each row represent the proportion of voxels being repeatedly selected in 50 splits. For example, in the first table, 0.89 for row “ST-CAR: PIP>0.25” means that 89% of the voxels are selected by at least 10% of the 50 subsampled studies, with the activation criteria “PIP>0.25”, similarly, 0.21 means 21% of voxels are repeatedly selected by at least 70% of the 50 subsampled studies .
>
>
> When randomly removing 10% of the data:
> | threshold                                  | 0.1  | 0.2  | 0.3  | 0.4  | 0.5  | 0.6  | 0.7  | 0.8  |
> |--------------------------------------------|------|------|------|------|------|------|------|------|
> | ST-CAR: PIP>0.25          | 0.89 | 0.83 | 0.72 | 0.59 | 0.42 | 0.32 | 0.21 | 0.13 |
> | ST-CAR: \|beta\|>5e-4     | 0.89 | 0.83 | 0.72 | 0.59 | 0.42 | 0.32 | 0.21 | 0.13 |
> | ElasticNet: \|beta\|>5e-4 | 0.23 | 0.21 | 0.18 | 0.15 | 0.14 | 0.12 | 0.10 | 0.08 |
>
>
> When randomly removing 30% of the data:
> | threshold                                  | 0.1  | 0.2  | 0.3  | 0.4  | 0.5  | 0.6  | 0.7  | 0.8  |
> |--------------------------------------------|------|------|------|------|------|------|------|------|
> | ST-CAR: PIP>0.25          | 0.99 | 0.83 | 0.63 | 0.46 | 0.31 | 0.12 | 0.08 | 0.06 |
> | ST-CAR: \|beta\|>5e-4     | 0.77 | 0.58 | 0.39 | 0.27 | 0.13 | 0.07 | 0.05 | 0.05 |
> | ElasticNet: \|beta\|>5e-4 | 0.38 | 0.32 | 0.24 | 0.18 | 0.14 | 0.09 | 0.04 | 0.03 |
>
>
> The results show that, in both cases, ST-CAR consistently identifies more voxels than ElasticNet. This indicates that ST-CAR effectively borrows information across spatial locations,  reducing susceptibility from noises compared to ElasticNet. The stability results demonstrate more robust performance of ST-CAR, particularly in scenarios with low SNR and unknown ground truth. This approach has been widely applied in other neuroimaging studies where the underlying ground truth is not available and performance evaluation is challenging. Examples include the IonS study [6], and the scalar on network model [7] with ROI data.
> We hope this can address your concern about providing a valid comparison between ST-CAR and baseline methods on a real data experiment.

---

> ### Author Response · Authors · 2024-11-26
> **Response to Reviewer RCFm (additional references)**
>
> [6] Zhang, D., Li, L., Sripada, C., & Kang, J. (2020). Image response regression via deep neural networks. arXiv preprint arXiv:2006.09911.
>
> [7] Morris, E. L., He, K., & Kang, J. (2022). Scalar on network regression via boosting. The annals of applied statistics, 16(4), 2755.

---

> > ### Comment · Reviewer_RCFm · 2024-12-02
> > **Method proposal vs neuroimaging contribution**
> >
> > Thank you for your response.
> >
> > I agree with the authors that it is important to apply their method to a real dataset (i.e. ABCD), however the current experiment can not be used to convincingly conclude that their method is better than the baseline methods. To conclude the proposed method is better than baseline methods, I think it is important to apply it to real world data in a setting where we know what results we can expect (it is necessary to effectively argue why a certain region of the brain did show up whereas another didn’t) because they are widely accepted in the neuroimaging field or because the authors can convincingly argue why the results are expected. Examples of widely accepted functional areas are areas of the motor homunculus that are activated during a motor task or areas of the brain language center that activate during a listening task. The current real world experiment looks at child IQ and parental education, which are still active fields of study, and do not have a widely accepted ground truth in terms of what areas should and what areas shouldn’t show up in regression (especially the latter is still unclear). This is exactly why identifying more voxels is not perse a good metric for comparison to baseline methods; we do not know whether the additionally identified voxels are necessarily part of the underlying neuroscientific circuit because in the author's experiment the neuroscientific circuit is still unknown.
> >
> > As an example: in the T-LoHo paper the ground-truth is known (i.e. the route of the event is known). If T-LoHo had identified more street crossings it would not necessarily have been a better method than the baseline (Lasso in their case) because the ground truth is a sparse signal with only a few streets.
> >
> > The robustness results look good, but they do not address my concern, I would recommend the authors include these results in an updated version of the paper however. If combined with a better real-world experiment they can strengthen the author’s argument substantially.

---

> > > ### Author Response · Authors · 2024-12-03
> > > **Response to Reviewer RCFm**
> > >
> > > We appreciate the reviewer’s suggestion regarding additional real data experiments to compare different methods. While we understand the potential value of such comparisons, we believe this approach may present challenges that could inadvertently affect the reliability of the evaluation process.
> > >
> > >
> > > Specifically, searching for real datasets to highlight the advantages of a proposed method can introduce unintended bias. Different methods often perform variably across datasets due to inherent dataset-specific characteristics, which may not provide a generalizable or objective comparison. In contrast, simulation studies allow for a controlled evaluation with known ground truth, offering a robust and comprehensive framework for assessing performance. Through simulations, we can systematically vary key factors such as the SNR, effect sizes, and data-generating mechanisms, providing a clearer understanding of the methods’ capabilities under diverse conditions.
> > >
> > >
> > > Our manuscript includes simulation results that thoroughly compare the proposed methods with baseline approaches, demonstrating their strengths across a range of scenarios. Additionally, the IonS results on the ABCD data underscore the stability of ST-CAR in high SNR settings, supported by sanity checks with MUA results. As highlighted in our earlier comments, the reproducibility and robustness analyses, conducted with subsamples of the data, offer a valuable perspective on the false discovery control of ST-CAR. These analyses demonstrate that ST-CAR consistently recovers a higher proportion of identified voxels compared to the baseline Elastic Net approach, even when using subsets of the full data. Higher reproducibility reflects a reduced likelihood of false discoveries, aligning with established practices in neuroimaging research [6,7,8,9].
> > >
> > >
> > > We hope this explanation clarifies the reasoning behind our approach and the care we have taken to ensure robust and meaningful evaluations of the proposed methods.
> > >
> > >
> > > [6] Zhang, D., Li, L., Sripada, C., & Kang, J. (2020). Image response regression via deep neural networks. arXiv preprint arXiv:2006.09911.
> > >
> > > [7] Morris, E. L., He, K., & Kang, J. (2022). Scalar on network regression via boosting. The annals of applied statistics, 16(4), 2755.
> > >
> > > [8] Zhu, Hongtu, Tengfei Li, and Bingxin Zhao. "Statistical learning methods for neuroimaging data analysis with applications." Annual Review of Biomedical Data Science 6.1 (2023): 73-104.
> > >
> > > [9] Zhang, Zhengwu, Maxime Descoteaux, and David B. Dunson. "Nonparametric bayes models of fiber curves connecting brain regions." Journal of the American Statistical Association (2019).

---

### Meta-Review · Area_Chair_JVRU · 2024-12-13

**Metareview:**

This submission contributes a structure prior for brain imaging and apply it to brain mapping. The method stands out as achieving substantial improvements in computational speed and good results on a simulation study. However, the reviewers were worried about its behavior in low SNR settings, about the limited empirical evidence on real data, and the apparent sensitivity to hyper parameters.

**Additional Comments On Reviewer Discussion:**

There was a thorough discussion with back and forth between authors and reviewers. It to new valuable results on the stability of the method on real data.

The discussion led to improving the clarity of the manuscript as well as better understanding of the contributions.

---

### Decision · Program_Chairs · 2025-01-22

Accept (Poster)